# Development of Genetically Encoded Fluorescent KSR1-Based Probes to Track Ceramides during Phagocytosis

**DOI:** 10.3390/ijms25052996

**Published:** 2024-03-05

**Authors:** Vladimir Girik, Larissa van Ek, Isabelle Dentand Quadri, Maral Azam, María Cruz Cobo, Marion Mandavit, Isabelle Riezman, Howard Riezman, Anne-Claude Gavin, Paula Nunes-Hasler

**Affiliations:** 1Department of Pathology and Immunology, Geneva Center for Inflammation Research, Faculty of Medicine, University of Geneva, 1211 Geneva, Switzerland; vlad.girik@gmail.com (V.G.); isabelle.dentandquadri@unige.ch (I.D.Q.); maral.azam@unige.ch (M.A.); maria.cruzcobo@unige.ch (M.C.C.); marion.mandavit@gmail.com (M.M.); 2Department of Cellular Physiology and Metabolism, Faculty of Medicine, University of Geneva, 1211 Geneva, Switzerland; larissa@lcvek.com (L.v.E.); anne-claude.gavin@unige.ch (A.-C.G.); 3Diabetes Center, Faculty of Medicine, University of Geneva, 1211 Geneva, Switzerland; 4Department of Biochemistry, NCCR Chemical Biology, Faculty of Science, University of Geneva, 1211 Geneva, Switzerland; isabelle.riezman@unige.ch (I.R.); howard.riezman@unige.ch (H.R.)

**Keywords:** kinase suppressor of ras 1 (KSR1), SET nuclear proto-oncogene, phosphatase 2A inhibitor 2 (I2PP2A), protein kinase c zeta (PRKCZ), C6-ceramide, sphingomyelinase, myriocin, fumonisin B1, stromal interaction molecule 1 (STIM1), Sec22b, lipidomics

## Abstract

Ceramides regulate phagocytosis; however, their exact function remains poorly understood. Here, we sought (1) to develop genetically encoded fluorescent tools for imaging ceramides, and (2) to use them to examine ceramide dynamics during phagocytosis. Fourteen enhanced green fluorescent protein (EGFP) fusion constructs based on four known ceramide-binding domains were generated and screened. While most constructs localized to the nucleus or cytosol, three based on the CA3 ceramide-binding domain of kinase suppressor of ras 1 (KSR1) localized to the plasma membrane or autolysosomes. C-terminally tagged CA3 with a vector-based (C-KSR) or glycine-serine linker (C-KSR-GS) responded sensitively and similarly to ceramide depletion and accumulation using a panel of ceramide modifying drugs, whereas N-terminally tagged CA3 (N-KSR) responded differently to a subset of treatments. Lipidomic and liposome microarray analysis suggested that, instead, N-KSR may preferentially bind glucosyl-ceramide. Additionally, the three probes showed distinct dynamics during phagocytosis. Despite partial autolysosomal degradation, C-KSR and C-KSR-GS accumulated at the plasma membrane during phagocytosis, whereas N-KSR did not. Moreover, the weak recruitment of C-KSR-GS to the endoplasmic reticulum and phagosomes was enhanced through overexpression of the endoplasmic reticulum proteins stromal interaction molecule 1 (STIM1) and Sec22b, and was more salient in dendritic cells. The data suggest these novel probes can be used to analyze sphingolipid dynamics and function in living cells.

## 1. Introduction

Phagocytosis is a fundamental process whereby external foreign particles are internalized by cells, and the resulting membrane-engulfed compartment, called the phagosome, undergoes a series of maturation steps that lead to the formation of the digestive milieu and degradation of the foreign material [1]. A large body of literature has highlighted the central role of dynamic phosphoinositide lipid remodeling in controlling this process through the recruitment of a variety of effectors including cytoskeletal, motor, or vesicular docking proteins that help guide phagosomes to their fate [1,2,3]. Sphingolipids are a diverse group of lipids crucial for maintaining the cellular membrane structure, fluidity, and curvature, that have long been recognized to additionally play bioactive signaling roles [4]. Ceramides, a central hub of the sphingolipid network [5,6], have also been implicated in various steps of the phagocytic process. These span from the facilitation of receptor clustering and activation [7,8], to the initial membrane deformation of phagocytic cups [9,10], to the promotion of phago–lysosome fusion [11,12,13], and the inhibition of antigen presentation in dendritic cells [14,15]. However, compared to phosphoinositides, less is known about the dynamics, regulation, and roles of ceramide and its related lipids during this critical immune process [16].

Arguably, part of the challenge may lie in the difficulty tracking these lipids in living cells [17]. Currently, mass spectrometry is considered the most reliable method for detecting and quantifying ceramides. However, it does not have the spatial resolution required for subcellular tracking [18] or relies on labor-, cost- and, material-intensive organelle isolation that still suffers from contamination of the endoplasmic reticulum (ER) membranes due to the tightly tethered membrane contact sites (MCSs) that may skew the lipid content [19]. In addition, tools such as fluorescent or functionalized sphingolipid analogs (including biotin, azido, or radiolabeled ceramides) are rapidly metabolized to other sphingolipid species and may partition in membranes in a different manner than their natural counterparts [17,20]. Moreover, although several anti-ceramide antibodies have been reported, such immunostainings are difficult since lipids are not directly fixed and may change locations during processing, and steps such as low temperature incubation and permeabilization necessary for the access of anti-ceramide antibodies to intracellular membranes, such as phagosomes, may disrupt natural lipid localization [17,21,22,23]. Therefore, an alternative approach enabling the detection of ceramides in living cells with high spatial resolution is desirable. Genetically encoded lipid probes, typically composed of lipid-binding domains coupled to a fluorescent tag, provide an opportunity to investigate the subcellular localization, relative cytosolic accessibility, and dynamics in living cells, and have been extensively employed to dissect phospholipid dynamics during phagocytosis [24]. A few described lipid probes specific for sphingolipids have been published [25,26]. However, no fluorescent ceramide biosensor based on isolated ceramide-binding domains have been described so far, although a range of full-length ceramide-binding proteins have been characterized [27,28,29,30]. Nevertheless, the localization of these proteins, which are all multi-domain proteins, may be governed by cooperative binding events, including ceramide, but also involving other factors or domains. An *Escherichia coli*-purified glutathione-S-transferase (GST) fusion of the CA3 ceramide-binding domain of the pseudokinase kinase suppressor of ras 1 (KSR1) was used in vitro and demonstrated ceramide-binding specificity [31], yet fluorescent versions of this construct for expression in mammalian cells have not been published. In the current report, we document the cloning and screening of 14 putative ceramide-binding genetically encoded fluorescent probes. Out of several potential ceramide-binding biosensors, three constructs all bearing the same CA3 domain of the human pseudokinase KSR1 were characterized further. These were named N-KSR, C-KSR, and C-KSR-GS according to which terminal (amino, N-, or carboxy, C-) the fluorescent protein was fused to, and the presence of a glycine–serine (GS)-rich linker. The three KSR1-based probes responded to treatments modifying the ceramide levels in cells, although changes in both the localization and global fluorescence levels differed between the three constructs depending on the stimulus. Immunostaining, liposome microarray, and lipidomic analyses, together with further testing of the cellular behavior of the probes, indicated that N-KSR may instead be more sensitive to glycosylated ceramides, C-KSR is less stable than the other two probes, and that both factors contributed to the differences in cellular responses observed. Interestingly, differential localization between the three probes, depending on the cell type and the expression of ER-localized constructs, was documented to occur during phagocytosis. Together, our data indicate that these constructs may be useful starting tools for dissecting subcellular sphingolipid dynamics and warrant further characterization.

## 2. Results

### 2.1. KSR1-Based Probes Associate to Cellular Membranes

In previous studies, ceramide has been reported to be present in multiple organelles, including the plasma membrane, ER, Golgi, mitochondria, lysosomes, and nucleus [22,32,33,34,35,36]. In artificial membranes, ceramide, a relatively hydrophobic lipid, readily flip-flops across bilayers [37,38]. However, whether mechanisms that actively maintain bilayer asymmetry of ceramides exist in cells, to our knowledge, has not been described. Thus, the localization pattern of a ceramide-binding probe could conceivably include binding to several different types of intracellular membranes. Many proteins, to date, have been identified as ceramide-binding proteins, including protein phosphatases (PP) PP1 and PP2A [39,40], cathepsin D [41], annexin A1 [42], and lysosome-associated protein transmembrane 4 (LAPTM4) [43]. However, in the search for potential ceramide-specific biosensors, we focused on proteins where the ceramide binding site had already been identified, and where this site was found in a protein domain that could be conceivably separated from the rest of the protein, with the hope of minimizing the protein’s natural activity or function. Thus, four domains that have been previously demonstrated to bind ceramides in vitro and/or in living cells were selected: (1) the earmuff domain (EMD) of the human nuclear proto-oncogene protein SET (amino acids (aa) 69–226) [44]; (2) the N-terminal C1 domain of atypical human protein kinase C zeta (PRKCZ, also called PKCζ), aa 123–193); (3) the C-terminal C20 domain (aa 405–592) of the same protein PRKCZ [32,45]; and (4) the CA3 domain (also called C1 or cysteine-rich domain, aa 317–400) of the human pseudokinase KSR1 [31] (Figure 1A). The selected domains were tagged with enhanced green fluorescent protein (EGFP) at the amino or carboxy terminus (prefixed with N- and C-, respectively) and expressed in mouse embryonic fibroblasts (MEFs). Fusion proteins based on SET (Figure 1B(i,ii)) and PRKCZ (Figure 1C(i–iv)) were primarily localized to the nucleus. A nuclear export signal (NES) was added to C-SET-EMD (Figure 1B(iii)) and C-PRKCZ-C20 (Figure 1C(v)) to release proteins trapped in the nucleus, a strategy previously used for other lipid biosensors [46]. C-NES-PRKCZ-C20 and C-NES-SET-EMD were mainly found in the cytosol, though nuclear localization was still noticeable (Figure 1B(iii),C(v)).

In contrast, the expression pattern of KSR1-based constructs depended on the position of the fluorescent tag and linker (Figure 1D). The N-terminally tagged KSR1-CA3 domain (N-KSR) was found primarily at the plasma membrane (Figure 1D(i)), while the C-terminally tagged probe (C-KSR) localized to large intracellular vesicles (Figure 1D(ii)). An examination of the vector-based linker in C-KSR revealed the introduction of a chaperone-mediated autophagy degradation signal RLELKLRILQ [47], and Western blot, as well as immunostainings with organelle markers transferrin, RAB7, and lysosomal associated membrane protein 1 (LAMP1), confirmed partial autolysosomal targeting and degradation (Appendix A). Targeted quantitative tandem mass spectrometry lipidomic analysis [48] showed that the total levels of glycerophospholipid species phosphatidylcholine (PC), phosphatidylethanolamine (PE), phosphatidylserine (PS), and phosphoinositide (PI) were unchanged. Measurements of the major sphingolipid classes, including ceramides, glycosylated ceramides (comprising glucosyl and galactosyl ceramides of identical mass, collectively termed hexosyl-ceramides), and sphingomyelins, revealed a small but significant increase in ceramides upon the overexpression of C-KSR (Appendix A). A closer inspection of individual lipid species showed that this was driven by a specific increase only in ceramides harboring a 16-carbon acyl chain (C16-ceramide, Appendix A). These results were consistent with a probe that is sensitive to ceramide but does not produce major global changes in sphingolipid metabolism. Furthermore, to circumvent autophagic destruction, a C-terminally tagged construct with an alternate glycine–serine (GS)-rich linker was generated and abbreviated C-KSR-GS (Figure 1D(iii)). This construct displayed a localization similar to N-KSR, although the plasma membrane localization was only salient in a subset of cells and was more stable than C-KSR (Figure 1D(i,iii) and Appendix A). To potentially increase the lipid biosensor affinity [24], a construct expressing two KSR1 CA3 domains in tandem fused at the C-terminus to EGFP was also generated (abbreviated 2x-C-KSR). However, the localization appeared similar to that of N-KSR and C-KSR-GS (Figure 1D(iv)). The localization of both N- and C-KSR constructs depended on two conserved cysteines (C359/C362) whose mutation to serine has previously been shown to decrease ceramide binding in vitro [31] (Figure 1D(v,vi)). This suggests that their subcellular location might indeed involve ceramide binding. When expressed in human Hela cells, probes based on KSR1, SET, and PRKCZ probes showed a subcellular localization pattern similar to that seen in MEFs (Appendix A), confirming the idea that they can function similarly in different cellular contexts. Since their expression pattern was suggestive of potential membrane binding, we then selected three of the KSR1-based probes for further characterization: N-KSR, C-KSR, and C-KSR-GS. Stable MEF cell lines for N-KSR and C-KSR were successfully generated and employed in subsequent experiments, while C-KSR-GS was characterized using transient transfection.

### 2.2. KSR-Based Probes Respond to Sphingolipid Depletion

Next, the behavior of the three KSR probes in response to changes in cellular ceramide levels was investigated. To induce ceramide depletion, cells were exposed to well-known inhibitors of sphingolipid biosynthesis: myriocin, a potent inhibitor of serine-palmitoyltransferase [49], the enzyme which catalyzes the first step in sphingolipid biosynthesis; and fumonisin B1, a selective inhibitor of ceramide synthases [50]. Upon ceramide depletion, C-KSR-expressing cells demonstrated a surprising and drastic 5-fold decrease in the intracellular fluorescence after a 3-day incubation with myriocin (0.5 µM) or fumonisin B1 (2.5 µM), accompanied by the expected re-localization to the cytosol, but only in a subset of cells (Figure 2A,B). In comparison, only a 2.5-fold decrease in the probe intensity was observed in N-KSR-expressing cells after sphingolipid depletion with either drug (Figure 2C,D). Targeted quantitative tandem mass spectrometry lipidomic analysis [48] confirmed that the total levels of all major sphingolipid classes measured, including ceramides, glycosylated ceramides (comprising glucosyl and galactosyl ceramides of identical mass, collectively termed hexosyl-ceramides), and sphingomyelins, were markedly depleted by the myriocin treatment (Figure 2E, see also Appendix A). In contrast, the total levels of the phospholipid species phosphatidylcholine (PC), phosphatidylethanolamine (PE), phosphatidylserine (PS), and phosphoinositide (PI) were unchanged (Figure 2E(i)). In addition, the myriocin treatment of cells transiently co-transfected with C-KSR and cytosolic Tag red fluorescent protein (TagRFP) did not produce any noticeable effect on TagRFP expression (Appendix A), demonstrating that the effect on the KSR construct expression was not due to a generalized effect on exogenous protein synthesis or degradation. In contrast, although no overt signs of toxicity were noted upon transient transfection of C-KSR-GS alone, a combination of 3-day treatment with myriocin or fumonisin B1 and C-KSR-GS transfection was toxic, precluding analysis, and indicating that the expression of this construct might sensitize cells to stress. However, an overnight treatment with 10 μM fumonisin B1 also resulted in a 2-fold decrease in the global cellular expression of C-KSR-GS (Figure 2F,G), indicating that, similar to both N-KSR and C-KSR, C-KSR-GS responds to sphingolipid depletion.

### 2.3. KSR-Based Probes React Differently to Ceramide Accumulation

To induce ceramide accumulation at different subcellular locations, two methods were employed. In the first, a treatment with a ceramide precursor, the 16-carbon acyl chain (C16) fatty acid palmitate for 4 h, was used to increase ceramide via de novo biosynthesis [51,52], which would be expected to increase the number of ceramides in the ER where biosynthesis takes place, as well as potentially in the secretory pathway [6]. In the second, cells were incubated for 30 min with the purified bacterial enzyme sphingomyelinase from *Staphylococcus aureus*. This enzyme directly cleaves sphingomyelin, a choline-ceramide derivative, to phosphorylcholine and ceramide [53,54]. Thus, it is expected to directly increase the number of ceramides on the outer leaflet of the plasma membrane while decreasing sphingomyelin, and it could potentially also increase the number of ceramides within the endocytic system [53,54]. The palmitate treatment (added complexed to bovine serum albumin (BSA), 0.5 mM) led to a 3-fold and 1.5-fold increase in global C-KSR and C-KSR-GS signal intensity, respectively, without an obvious shift in subcellular localization as compared to control cells treated with BSA alone (Figure 3A,B). This contrasted with the N-KSR probe which did not show any changes in fluorescence intensity or localization (Figure 3C). The lipidomic analysis showed that the palmitate treatment induced a 20-fold accumulation of total ceramides, driven mainly by the accumulation of C16-ceramide as expected (Figure 3D(i–iii), see also Appendix A). On the other hand, palmitate did not induce significant changes in total PC, PE, PI, or PS, nor in hexosyl-ceramides or sphingomyelins, although a trend for increased hexosyl-ceramides and decreased sphingomyelins was noted. (Figure 3D). However, upon closer inspection of individual lipid subspecies, the only significantly increased hexosyl-ceramide was that of C10-hexosyl-ceramide, a lipid of unknown biological significance. Unexpectedly, a significant decrease in C24-sphingomyelin, but not other sphingomyelin subspecies, was also noted (Figure 3D(iii)).

Next, the two C-terminally tagged constructs C-KSR and C-KSR-GS, which showed only partial plasma membrane localization under resting conditions, were examined for their response to treatment with purified bacterial sphingomyelinase (0.5 U/mL dissolved in PBS, diluted in serum-free medium) for 30 min (Figure 4A). This treatment, expected to convert the outer leaflet plasma membrane sphingomyelin into ceramide [53,54], led to a small but significant increase in the global C-KSR fluorescence intensity compared to controls treated with PBS in a serum-free medium (Figure 4B). In addition, it also led to a 2-fold increase (from 29 to 59%) in the recruitment of C-KSR to the plasma membrane, measured as the fraction of cells displaying plasma membrane localization in sphingomyelinase-treated cells as compared to controls (Figure 4A insets, Figure 4C). A significant 1.5-fold increase in the fraction of cells displaying plasma membrane localization of C-KSR-GS (from 35 to 54%) was also observed (Figure 4D,E), whereas an analysis of the N-KSR was not conducted since nearly all cells displayed plasma membrane localization at resting conditions. A lipidomic analysis revealed that sphingomyelinase significantly increased the total ceramides 2-fold and decreased the total sphingomyelins by 40%, whereas levels of total hexosyl-ceramides, PC, PE, PI, and PS were unchanged (Figure 4F(i)). The increase in ceramides was largely driven by increased C16-ceramide, with C18- and C22-ceramides also contributing (Figure 4F(ii), see also Appendix A). Interestingly, C24-ceramide was slightly but significantly decreased, whereas the decrease in sphingomyelin was driven by the decreased C18-, C22-, and C24-, but not C16-sphingomyelin (Figure 4F(iii)). These data suggest that this bacterial sphingomyelinase displays a preference for cleavage of sphingomyelin species harboring longer (18- to 24-carbon) acyl chains and that the cleavage generates C18- and C22-ceramide. The greater generation of C16-ceramide may thus result from the conversion of C24- to C16-ceramide, but might also represent C16-ceramide generation through a different pathway. In addition to the lipidomic data indicating the efficient cleavage of sphingomyelins, levels of sphingomyelin were additionally assessed by fixing cells after the sphingomyelinase treatment and staining them with a purified EGFP-tagged sphingomyelin-specific probe lysenin (20 µg/mL) [55]. Lysenin staining was lost upon sphingomyelinase treatment, confirming efficient cleavage of plasma membrane sphingomyelin under these conditions (Appendix A).

Diacylglycerol (DAG) is a molecule with a close structural similarity to ceramide that could potentially be indirectly generated through the activation of SM synthases upon large increases in ceramide, such as those observed under the sphingomyelinase treatment here [56,57]. Thus, the specificity of C-KSR for ceramide was further investigated through the generation of C-KSR-mRFP and co-expression with the DAG probe GFP-PKC-C1(2) [58]. Cells were then treated with either 10 µM ionomycin (Iono) for 10 min in order to induce plasma membrane DAG accumulation [59], or with sphingomyelinase as above. The ionomycin treatment, but not the sphingomyelinase, induced PKC-C1(2) plasma membrane recruitment, while the opposite pattern was observed for C-KSR (Figure 4G,H). The treatment with sphingomyelinase did not affect the localization of PKC-C1(2) when it was expressed alone, nor did it affect the localization of an alternate DAG probe YFP-DBD [60] (Appendix A), ruling out potential indirect DAG accumulation after treatment with sphingomyelinase. Collectively, the data suggested that, out of the three probes analyzed, C-KSR, despite the drawback of autolysosomal degradation, is the most sensitive to the changes in cellular levels of ceramides and can also detect locally generated ceramides at the plasma membrane. In addition, the data showed that C-KSR-GS responds similarly to C-KSR but with a slightly smaller dynamic range. Importantly, however, the lack of response of N-KSR to palmitate suggests a potential difference in lipid binding specificity.

### 2.4. N-KSR Binds Glucosyl-Ceramide In Vitro

To further investigate the mechanism underlying the differences in how the KSR-based probes react to ceramide accumulation, we sought to measure their in vitro binding specificity to artificial membranes containing different ceramide species and derivatives. To this end, a liposome microarray-based assay (LiMA) was employed, where arrays of liposomes (small artificial vesicles) of various compositions are deposited inside a microfluidic chip, exposed to purified (or cell lysates containing) fluorescent lipid-binding probes, and binding is assessed using automated fluorescence microscopy [61]. To obtain purified probes, the N-KSR and C-KSR probes were re-cloned into super-folding GFP (sfGFP) bacterial expression vectors [62] and purified from *E. coli* cultures. Interestingly, significant binding of N-KSR was observed to dioleoyl-phosphatidylcholine (DOPC) liposomes containing 2% glucosyl-ceramide (GlcCer), whereas the trends toward binding to liposomes containing either 10% sphingomyelin or 10% bovine ceramide mixtures were not significant (Figure 5A and Appendix A). Purified C-KSR tended to aggregate in this assay, and the sample sizes were too small (less than *n* = 3) to allow statistical analysis on individual types of membrane spots. However, pooling the binding scores of control liposomes (lacking ceramide) and comparing this to pooled scores of liposomes containing ceramide or ceramide derivatives (lipids of close structural similarity to ceramide) suggests a preference for membranes harboring ceramides or ceramide-like lipids (Figure 5B). We also studied the binding of C-KSR-GS expressed in lysates from transfected HEK293 cells, to ceramide present in liposomes with either DOPC as the carrier lipid as above, or in liposomes based on palmitoyl-oleoyl-PC (POPC) that mimic the composition of the inner plasma membrane (IPM liposomes) [63] (Figure 5C and Appendix A). Lysates from HEK cells expressing soluble EGFP and the DAG probe PKC-C1(2) were employed as negative and positive controls, respectively (Appendix A). Whereas robust and significant binding of PKC-C1(2) to DAG-containing liposomes as compared to IPM liposomes was readily observed (Appendix A), the trend for a difference in binding scores of C-KSR-GS to individual ceramide containing liposome compositions as compared to controls (DOPC, POPC or IPM only) alone were not significant (Appendix A). On the other hand, when pooling liposome data to increase statistical power as above, a comparison of pooled control liposome spots (DOPC, POPC and IPM alone) to DOPC or IPM liposomes containing ceramides revealed a small but significant difference in binding scores in the IPM context (Figure 5C). These data support a preference of C-KSR-GS for ceramide-containing membranes within a more complex lipid context.

Since the in vitro conditions tested likely do not fully recapitulate the optimal conditions for lipid binding of our probes, we continued our exploration of the differences between C- and N-terminally tagged probes within the cellular context. Thus, we next examined the response of the two probes displaying the most similar localization pattern (N-KSR and C-KSR-GS) to additional reagents previously shown to increase cellular ceramides. To this end, cells expressing either probe were treated for 2 h with 200 μM of the ceramide analogue C6-ceramide [64] or overnight with 200 μM of the glucosyl-ceramide synthase inhibitor Genz-123346 [65] or their respective ethanol and methanol vehicle controls. Previously, C6-ceramide has been shown to induce large increases in endogenous ceramides and hexosyl-ceramides, with an only comparatively smaller increase in sphingomyelin [64,66,67]. In contrast, Genz-123346 increased the ceramides without affecting sphingomyelin and concomitantly decreasing hexosyl-ceramides [65,68]. Interestingly, whereas both probes showed a robust increase in cellular fluorescence upon treatment with C6-ceramide (Figure 5D–G), a strikingly different response to Genz-123346 treatment was noted, where an increase in fluorescence was observed with C-KSR-GS but not with N-KSR (Figure 5D–G). Together, these data suggest that a preference of N-KSR for glucosyl-ceramide instead of ceramide could explain the difference in cellular behavior of the N- and C-terminally tagged probes.

### 2.5. KSR1-Based Probes Behave Differently during Phagocytosis

In several reports, ceramide generation has been associated with phagocytic ingestion and phagosomal maturation, and has been shown to be important for pathogen elimination, yet its precise role is still poorly understood [69,70]. Thus, to not only glean some hints at ceramide dynamics (and eventually function) during this important immune process, but also to test our probes under a physiological stimulus, we next investigated the behavior of our three KSR1-based probes during phagocytosis. To this end, we examined MEFs rendered phagocytic through overexpression of the Fc gamma receptor IIa (FCGR2A)-c-myc. This phagocytic cell model is easier to transfect and manipulate than naturally phagocytic cells, and we and others employing a similar strategy have observed such models recapitulate many aspects of the phagocytic process, including phospholipid dynamics [13,71,72,73]. Cells expressing each of the three KSR1-based probes (C-, N- and C-KSR-GS) were exposed to 3 µm IgG-opsonized polystyrene beads for 30 min or 135 min, time-points representative of the early and late stages of phagosomal maturation (Figure 6A–C). In parallel, sphingolipid levels were measured in C-KSR-expressing cells using targeted lipidomics, where a large accumulation in total cellular ceramides, sphingomyelin, and hexosyl-ceramide (4-, 2.6-, and 6-fold, respectively) was observed at 30 min (Figure 6D, see also Appendix A). This effect was driven by changes in C16- and C24-ceramide, C24-sphingomyelin, and C16- and C24-hexosyl-ceramide species (Figure 6D(i,ii)). It should be noted that changes in the total PC, PI, and PS were also observed (Figure 6D(iii–iv)). Interestingly, 30 min after the incubation with IgG-coupled beads, an accumulation of C-KSR at the plasma membrane, similar to what was observed after incubation with bacterial sphingomyelinase (and blindly quantified in a similar manner), was noted (Figure 6A, see also Figure 4A–C,G–H). Exploring this plasma membrane localization further, we found that plasma membrane recruitment of C-KSR could still be detected at 45 min and 60 min after phagocytosis initiation, yet was mostly lost at later stages, after 90 and 135 min, correlating closely with changes in the sphingolipid levels at 30 and 135 min (Appendix A and Figure 6A,D). For N-KSR and C-KSR-GS, the FCGR2A expression appeared to increase plasma membrane localization already at baseline, precluding the simplified binary assessment of plasma membrane recruitment as performed for C-KSR. Thus, instead, the plasma membrane and total cellular fluorescence intensity levels (excluding regions containing phagosomes) were quantified using artificial intelligence-assisted image segmentation. Interestingly, a loss of total fluorescence was noted for C-KSR-GS at 135 min, whereas the trend for a similar loss was not significant for N-KSR (Figure 6B,C). In contrast, the plasma membrane appeared to be more salient upon phagocytosis for both constructs. This was detected by normalizing the plasma membrane to total cell fluorescence, where this ratio was significantly increased at both 30 and 135 min for C-KSR-GS, and only at 135 min for N-KSR (Figure 6B,C). Moreover, whereas N-KSR was completely absent from phagosomes (Figure 6C, insets), in C-KSR-GS-expressing cells, an accumulation around phagosomes was detected in a subset of phagosomes (Figure 6B, arrows). In some instances, a pattern of small, flattened cisternae or protruding tubular structures could be seen around phagosomes, which were reminiscent of membrane contact sites (MCS) between the phagosomal membrane and ER, structures that have been studied extensively by our laboratory in the past [73,74,75,76] (Figure 6B, arrows). To confirm whether such structures could indeed represent ER, C-KSR-GS was then co-expressed with several ER-targeted constructs, including a soluble ER-targeted RFP TagRFP-KDEL, a calcium regulator and MCS marker monomeric Cherry (mCh)-STIM1 and mCh-Sec22b, a protein regulating traffic between ER and Golgi that we have recently shown to tether STIM1-independent MCS [73,75] (Figure 7A,B and Appendix A). Interestingly, although an overlap between C-KSR-GS and KDEL was not easily discerned, likely due to the low overall KDEL expression (Appendix A), an overlap between C-KSR-GS and STIM1, as well as C-KSR-GS and Sec22b, was observed in some but not all instances of periphagosomal accumulation of the fluorescent proteins (Figure 7A,B). Finally, we expressed the N-KSR and C-KSR-GS probes in a naturally phagocytic dendritic cell line, JAWS II. Similar to MEFs, N-KSR did not localize to the phagosomal or other organellar membranes (Figure 7C). In contrast, a robust C-KSR-GS accumulation at the phagosomal membrane could be discerned on a subset of phagosomes, as well as on internal structures reminiscent of the ER (Figure 7D).

Several inferences may be gleaned from these intriguing observations. First, the large changes in ceramides during the first 30 min of phagocytosis detected in whole-cell MEF extracts were likely occurring at the plasma membrane and were best detected using C-KSR, which is the most sensitive of the three probes at detecting dynamic changes of plasma membrane ceramides, though C-KSR-GS is also capable of detecting changes in plasma membrane ceramides. The fact that the C-KSR-GS signal is specific is also highlighted by the observation that the highly similar N-KSR construct did not behave in a similar manner. Indeed, whether N-KSR is truly capable of sensing glucosyl-ceramide or other ceramide derivatives in the cellular context remains to be verified, yet clearly it is not sensing ceramide itself, at least not to the same extent as C-KSR-GS, and maybe a useful control to show specificity for studies employing C-KSR-GS. However, the fact that the relative C-KSR-GS levels at the plasma membrane remained high at 135 min was surprising, as was the time-dependent global loss in total cellular fluorescence. Whether C-KSR-GS expression itself impacts ceramide dynamics, for example, by delaying the ceramide loss at later time points, or whether ceramide-independent effects on C-KSR-GS degradation can account for these observations also remain to be defined. On the other hand, C-KSR-GS may be additionally useful in detecting ceramide species on intracellular membranes, including phagosomal and ER membranes, since the data also hint at dynamic ceramide fluxes, potentially at contact sites between the phagosomal membrane and ER.

## 3. Discussion

Despite several proteins reported as ceramide binders [17,28,30], to our knowledge no genetically encoded ceramide-specific probes per se have been reported. Although whole GFP-tagged ceramide-binding proteins were employed to track subcellular ceramide in a limited number of studies, they contain other active domains that could contribute to the signal. Out of the 14 candidate biosensor constructs reported here, we focused on three based on the KSR1 CA3 (317–400) ceramide-binding domain which displayed membrane localization when expressed in cells. In contrast, constructs based on PRKCZ and SET were targeted to the nucleus, without an obvious association to cellular membranes. Since sphingolipids have been detected in both the nuclear membrane as well as the nuclear matrix [35], and SET has a defined nuclear activity [36], these proteins may detect ceramides in the nuclear matrix rather than simply being misfolded [77]. Future studies employing stimuli such as DNA damage, known to change nuclear ceramides, may yet verify the utility of these constructs that were not further characterized here.

In contrast, though nuclear shuttling has also been reported for KSR1, a 934-amino acid scaffolding regulator of the rat sarcoma virus small GTPase/rapidly accelerated fibrosarcoma/mitogen-activated protein kinase kinase/extracellular signal-regulated kinases pathway (referred to as Ras/Raf/MEK/ERK signaling) comprised of five domains (CA1–CA5) [78], KSR1 normally resides in the cytosol in resting conditions [79]. It is recruited to the plasma membrane upon growth factor stimulation, which induces both the activities of ceramide-generating sphingomyelinases, as well as changes in the phosphorylation of multiple KSR1 residues [80,81]. Several studies have reported direct ceramide binding of the KSR1 CA3, including in vitro reports using an N-terminal GST fusion with a small CA3 fragment of similar composition (aa 320–388) as the one employed here (aa 317–400), though others, notably some employing a truncated CA3 (aa 334–377), have failed to detect direct ceramide binding [31,82,83,84]. Thus, we hypothesized that probes based on this small 83 amino acid fragment of KSR1 are particularly poised to detect plasma membrane ceramides, while reducing the likelihood that the protein’s natural function or other binding partners will interfere with its localization and behavior. In addition, since the MEK and ERK interaction domains (CA4–CA5) are absent, the oncogenic potential of a construct derived from KSR1 [78] would also be expected to be circumvented, mitigating safety concerns for the use of these constructs. However, mutagenesis studies suggest that the phosphorylation of S392, a residue found within the CA3 domain, contributes to KSR1’s stability and membrane re-localization upon growth factor activation [79,81]. While full-length KSR1-S392A localizes to the nucleus rather than the plasma membrane [79], whether the phosphorylation status of S392 in the isolated CA3 of our KSR constructs contributes to membrane localization remains to be defined.

In the characterization of our probes, a surprising observation was that the most salient change in the behavior of the KSR probes in response to ceramide-modifying drugs was a change in the total cellular fluorescence. This contrasts with similar phospholipid probes, which usually change their locations rather than their total fluorescence. The effect was consistently observed in both the induction and the depletion of ceramides and across all drug treatments tested (Figure 2, Figure 3, Figure 4 and Figure 5), and at least for C-KSR, these changes correlated well with levels of ceramide measured using mass spectrometry for all conditions tested. For the myriocin treatment, a simple toxicity effect (which often accompanies changing levels of ceramide [6]) on the global exogenous gene expression was ruled out since levels of cytosolic RFP, expressed from a similar cytomegalovirus (CMV) promoter-containing plasmid, were unchanged. In addition, with the sphingomyelinase treatment, neither of the two DAG probes reacted in this manner (Figure 4 and Appendix A). Why should a lipid probe change its global fluorescence in response to lipid changes? A potential explanation is that the CA3 domain itself is somewhat unstable and subject to an intrinsic degradation that is mitigated by membrane binding. Indeed, full-length KSR1 binds cytosolic chaperones heat-shock proteins (Hsp)70, 90, and 68, [85], though the binding sites are not defined, and KSR1 is ubiquitinated, though at a site outside the CA3 domain [86,87]. Moreover, phosphorylation at S392 (together with T274) increases the half-life of KSR1 from 1 h to 6 h [87]. Such a fast turnover time is consistent with our results, which were observed even with treatments as short as 30 min. However, whether phosphorylation, ubiquitination, or chaperone binding might also be modified, increasing the stability of the cytosolic fraction of the probe will also be important to define in future studies. Indeed, a careful analysis of changes in probe abundance, monitored using both Western blot and fluorescence, to inhibitors of lysosomal or proteasomal degradation, small interfering ribonucleic acid (siRNA)/ clustered regularly interspaced short palindromic repeats (CRISPR) against chaperones, and the generation of S392 mutations will be greatly informative to dissect the influence of probe stability on the probe’s response to changes in cellular ceramide. A more detailed characterization of the probe stability will also be important to define whether these probes are suitable to longer term imaging studies.

In addition to its potential influence on cellular probe behavior, an intrinsic instability of the CA3 domain could well explain the weak in vitro binding observed for C-KSR-GS and C-KSR to artificial membranes containing ceramides or ceramide derivates. Thus, while the precise specificity and selectivity of the probes for ceramides remain to be more firmly established, the propensity for the isolated probes to aggregate and precipitate in solution have rendered in vitro studies, which are usually employed to define this (such as liposome flotation or fat blot), impractical. Thus, future studies directed at dissecting the specificity and selectivity of these probes within the cellular context will likely be more fruitful than biochemical exploration.

Nevertheless, a significant and surprising interaction of the N-KSR probe to glucosyl-ceramide-containing liposomes was observed. Though unexpected, such a modification to the lipid specificity of the CA3 domain is consistent with the clear difference in behavior between N and C constructs to palmitate, Genz-123346, and phagocytic stimuli. Why should the position of the fluorescent protein tag have such an influence on the lipid binding properties of the KSR1 CA3 domain? The structure of a truncated CA3 (aa 334–377) indicated a shallow putative ceramide binding pocket involving residues L342, V345, M353, I354, and F355 [82]. Thus, such shallow interaction surface could conceivably be deformed by N-terminal attachment to a rigid or bulky domain. A wider pocket might thus decrease the original affinity of the CA3 for ceramides and instead favor binding ceramide species bearing a small additional head-group such as a single sugar or phosphate group. However, during phagocytosis the behavior of N-KSR did not correlate with hexosyl-ceramide levels measured using mass spectrometry. Though several potential factors may explain this, including the fact that changes in glucosyl-ceramide might not occur on the cytosolic leaflet, since flip-flop is not efficient for glycosylated ceramides [38,88,89,90], we cannot exclude that the N-KSR binding profile within the cellular context differs from its in vitro profile. Thus, future studies with drugs or the genetic manipulation of enzymes, such as glucosyl and galactosyl ceramidases, ceramide kinase, or transporters, such as anoctamin 6 (ANO6, also called TMEM16F) [88] and ceramide transferase, will be useful to corroborate these findings and determine more precisely the specificity and selectivity of the three probes for glycosylated or phosphorylated ceramides versus ceramide itself in the cellular context. In addition, performing calibration experiments where a range of cellular ceramide concentrations, verified using mass spectrometry, are directly correlated to levels of C-KSR or C-KSR-GS responses will be extremely useful in establishing the true dynamic range and ceramide sensitivity and utility of these probes.

Overall, the examination of our three KSR1-based probes under different stimuli (summarized in Figure 8) demonstrated that these probes may each have their own utility as such, or serve as a basis for future designs and studies aimed at dissecting ceramide (and potentially glucosyl-ceramide) dynamics in cells. The exacerbated instability and autolysosomal degradation of C-KSR is certainly a limitation, since a contribution of a partially degraded probe to the fluorescent signal (that an intact fluorescent protein separated from the lipid binding domain might generate) cannot be excluded to contribute to the signal of internal membranes. On the other hand, this very instability, which likely explains the smaller constitutive plasma membrane localization, proved in fact to be advantageous in the way that it allowed for a greater dynamic range and more salient plasma membrane recruitment with a phagocytic stimulus than C-KSR-GS. Indeed, C-KSR expression had only a minimal effect on the global levels of sphingolipids (Appendix A), and more closely followed the measured levels of ceramide during phagocytosis (Figure 6A) than C-KSR-GS. Combined with the cell loss under extended myriocin treatment, and the difficulty in establishing a stable cell line, C-KSR-GS expression likely exerts a certain level of toxicity. Thus, close monitoring for interference with regular cellular function, mis-localization or aggregation depending on the manipulation, as well as long-term stability, are all factors that should be carefully considered for future applications using these probes. Thus, similar to the conditional instability employed in nanobody engineering [91], a future design could build upon these observations to include proteasome targeting or other degradative signals, which may be less disruptive than the current design to membrane homeostasis [92], to obtain a sensitive on–off biosensor for plasma membrane ceramide.

Finally, as mentioned above, other methods currently employed to track subcellular ceramides, such as ceramide-specific antibodies and fluorescent or functionalized ceramide analogs, each have their drawbacks, such as the technical difficulties of lipid immunostaining and rapid metabolism, or the altered localization of modified lipids [17]. Thus, despite the limitations, the KSR-based probes characterized in the present study will surely provide novel insights into ceramide and sphingolipid cellular dynamics, particularly if several corroborating methods are compared. Indeed, the sphingomyelin probe lysenin has recently contributed greatly to our understanding of the role of sphingomyelin (and indirectly ceramide) in membrane repair and antigen cytosol transfer in dendritic cells [93,94,95], and future studies analyzing the behavior of the KSR-based probes to membrane-disrupting stimuli may reveal new insights in the role of ceramide in membrane repair [96] and ceramide-mediated cell death [97]. Moreover, in addition to the role of KSR1 itself to RAS-mediated oncogenesis and its potential as a novel anti-cancer target [78], sphingolipid dysregulation has been linked to several cardiovascular, metabolic, and neurodegenerative diseases [98,99]. Thus, the KSR-based probes described here could contribute not only to a greater understanding of the pathophysiology underlying these diseases, but they may also help identify and characterize novel compounds and methods for future diagnostic or therapeutic clinical use. Although clearly more work will be required to define the behavior and lipid specificity of these novel biosensors, we believe these constructs represent useful new tools that complement and extend the existing tools for the exploration of sphingolipid regulation and function in living cells.

## 4. Materials and Methods

### 4.1. Reagents

The following antibodies (antibody Name/catalog number/dilution) were from Abcam (Cambridge, UK)—anti-RAB7/ab137029/1:100; Enzo Life Sciences (Farmingdale, NY, USA)—anti-LAMP1/ADI-VAM-EN001/1:50; Thermo Fisher Scientific (Thermo, Waltham, MA, USA)—AlexaFluor-(AF)555 Goat anti-mouse IgG/A-21422/1:500, AF555-goat-anti-rabbit IgG/A-21428/1:250; Santa-Cruz Biotechnology (Dallas, TX, USA)—Goat anti-rabbit IgG-HRP/sc-2030/1:5000, mouse anti-GFP (B-2)/sc-9996/1:1000, mouse anti-tubulin (DM1A)/sc-32293/1:5000; Bio-Rad Laboratories (Hercules, CA, USA)—Goat anti-mouse IgG-HRP/170-6516/1:3000. All lipid standards for lipidomic analysis were from Avanti Polar Lipids (Alabaster, AL, USA), with amounts given per 6 cm cell culture dish: DLPC (12:0-12:0 PC, 0.4 nmol/dish, cat. #850335), PE31:1 (17:0-14:1 PE, 1 nmol/dish, cat # LM-1104), PI31:1 (17:0-14:1 PI, 1 nmol/dish, cat # LM-1504), PS31:1 (17:0-14:1 PS, 3.3 nmol/dish, cat# LM-1304), CL56:0 (14:0 Cardiolipin, 0.7 nmol/dish, cat#710332), C12SM (C12 Sphingomyelin d18:1-12:0, 0.1 nmol/dish, cat#860583), C17Cer (C17 Ceramide, d18:1-17:0, 2.5 nmol/dish, cat #860517), and C8GC (C8 Glucosyl(B)Ceramide, d18:1-8:0, 0.5 nmol/dish, cat#860540). Lipids employed in the liposome microarray analysis are listed in Appendix A. Chemicals were purchased from MilliporeSigma (Burlington, MA, USA) unless otherwise stated, (Name/catalog #): HPLC grade Methanol/4860-1L-R, HPLC grade Chloroform/650471, tert-Butyl Methyl Ether (MTBE)/650560, Methylamine/534102; LC-MS grade water/1153331000. Thermo: HPLC grade n-butanol/11378197. Plasmids used and generated in this study are listed in Appendix A.

### 4.2. DNA Cloning

All newly made constructs were verified using sequencing. Constructs expressing candidate ceramide probes were constructed by subcloning the human KSR1 CA3 domain (aa 317–400, XM_047436985), N-terminal C1 domain of human PRKCZ (aa 123–193, NM_002744.6), the C-terminal C20 domain of human PRKCZ (aa 405–646), or human SET EMD domain (aa 70–226, NM_001122821.2) into pEGFP-N1 or pmRFP-N1 using restriction sites NheI and XhoI, or into pEGFP-C1 using XhoI and KpnI sites to create C- or N-terminally tagged proteins, respectively. The C-KSR-GS construct replacing the vector-based linker (LELKLRILQS) with a glycine-serine rich linker (GGSSGGGGA) was generated using site-directed mutagenesis (Q5 Site-Directed Mutagenesis Kit, New England Biolabs, Ipswich, MA, USA) of the C-KSR-EGFP plasmid. The tandem 2x-C-KSR probe was made by subcloning a second CA3 domain together with a flexible linker sequence (GGSSGGGGA) introduced in the primer between the two CA3 domains into C-KSR-EGFP using XhoI and KpnI sites. The sequence encoding the nuclear export signal (NES) from mammalian MAPKK kinase (LQKKLEELEL) was inserted using site-directed mutagenesis after the start codons of human PRKCZ C20(405–646) or SET EMD(70–226) to yield plasmids containing C-NES-C20-PRKCZ-EGFP and C-NES-SET-EMD-EGFP, respectively. Two point mutations in the CA3 domain of KSR1 (C359S, C362S) with substitution of the two conserved cysteines with serine residues were introduced using the Q5 Site-Directed Mutagenesis Kit. For protein purification from *E. coli*, the KSR1 CA3 domain was cloned into SfiI restriction sites of the pETM11-SUMO3-sfGFP [61] to make pETM11-SUMO3-sfGFP-KSR1-CA3. The sequence encoding C-KSR was subcloned into pETM11-SUMO3 via Gibson assembly using Gibson Assembly Cloning Kit (New England Biolabs). Sequences encoding N-KSR or C-KSR were subcloned into SfiI sites of pSBbi-pur-H-2Kb (# 111623, Addgene, Watertown, MA, USA), replacing an existing H-2kb fragment. Primers used are listed in Appendix A.

### 4.3. Cell Culture, Transfection, and Stable Cell Line Establishment

Wild-type mouse embryonic fibroblasts (MEF, ATCC, Manassas, VA, USA; CRL-2991), HeLa cells (European Collection of Authenticated Cell Cultures, Salibury, UK; 93021013, a gift from N. Demaurex), and HEK 293T (HEK, ATCC, CRL-11268, a gift from N. Demaurex) were cultured in DMEM (22320, Thermo), MEM (41090, Thermo) and DMEM glutaMAX (31966, Thermo), respectively, containing 10% fetal bovine serum (FBS, Thermo) and 1% pen/strep (10000 U/mL penicillin, 10 mg/mL streptomycin, PAN Biotech, Aidenbach, Germany). JAWS II (JAWS, ATCC CRL-11904) was grown in alpha minimum essential medium with ribonucleosides and deoxyribonucleosides (Thermo, 22571020) supplemented with 4 mM L-glutamine (Thermo, 25030149), 1 mM sodium pyruvate (Thermo, 11360070), 5 ng/mL GM-CSF (Peprotech, London, UK, 315-03) 20% FBS, 1% pen/strep. Delipidated FBS was prepared by incubating FBS with 0.5 g/mL lipid removal beads (13358-U, MilliporeSigma) at 4 °C overnight with rotation followed by centrifugation (27,000× *g*, 20 min, 4 °C) and supernatant filtration using 0.2 μm Nalgene PES filters (Thermo, 725-2520). All cell lines were cultured at 37 °C, 5% CO_2_, and passaged 1–2 times per week, and used between passages 10 and 40. All cell lines were tested every 6–12 months either using PCR (LookOut kit, MilliporeSigma) or Mycostrips (Invivogen, San Diego, CA, USA, ep-mysnc-50) and were negative. Cells were transfected upon reaching 60–70% confluency 2–3 days after seeding using Lipofectamine 2000 (Thermo, 11668500) in complete culture medium without antibiotics for 4–6 h. To prepare DNA–lipofectamine complexes, equal volumes of serum- and antibiotic-free media containing DNA or Lipofectamine were mixed and incubated at room temperature for 1 h. To obtain stable cell lines via transposition, MEF cells were first co-transfected with pSBbi-pur plasmids encoding the protein of interest and pCMV-(CAT)T7-SB100 plasmid (#34879, Addgene) encoding Sleeping Beauty transposase SB100x. Puromycin (10 μg/mL) selection was performed 2 days after co-transfection. Stable cell lines (N-KSR and C-KSR) were maintained in complete culture medium containing puromycin (10 µg/mL). JAWS transfection was achieved using 20 μg DNA and 2 × 10^6^ million cells per 100 μL nucleofection solution from the Amaxa Nucleofector Mouse Dendritic Cell kit (Lonza Group, Basel, Switzerland), and the Y-001 program. Cells were diluted in 400 μL RPMI 1640 Glutamax medium and 25 mM HEPES medium (Thermo, 72400021) immediately after electroporation, incubated for 10 min at RT, then centrifuged at 200× *g* for 5 min at RT and resuspended in warm complete medium and incubated overnight before imaging.

### 4.4. Immunofluorescence

MEF and HeLa cells seeded on 0.17 mm glass coverslips (Carl Roth, Karlsruhe, Germany) were fixed with 4% paraformaldehyde (PFA, AlfaAesar, WardHill, MA, USA) in PBS for 30 min at RT, followed by three washes in PBS. JAWS were fixed in 1%PFA for 10 min. For staining nuclei, fixed cells were incubated with 10 µg/mL Hoechst 33342 (Thermo, 62249) for 5 min, followed by three washes with PBS and mounting. For immunostaining, cells were permeabilized with 0.1% Triton X-100 (in PBS) for 5 min at RT, washed three times using PBS, quenched with 25 mM glycine/PBS for 10 min at RT, and treated with Image-IT-FX (Thermo, I36933) for 30 min, followed by blocking with 2% bovine serum albumin (BSA)/PBS for 30 min. After that, the cells were incubated with primary antibodies at 4 °C overnight and with a secondary antibody for 1 h in dark humid chambers, antibodies diluted in blocking buffer. Three washes in PBS were carried out between each step in the procedure. Finally, the coverslips were mounted onto glass slides using 5 µL of ProLong Diamond Mounting medium (Thermo, P36970) per coverslip. For labeling recycling endosomes, MEF cells were incubated for 30 min at 37 °C with 50 µg/mL AF555-conjugated human transferrin (Thermo, T35352) in MEF complete culture medium containing 0.2% (BSA). After two washes with PBS, cells were fixed with 3% PFA for 20 min, quenched with 50 mM ammonium chloride for 15 min (RT), followed by 3 washes in PBS and mounting. Images were acquired either using Zeiss (Oberkochen, Germany) LSM700 or LSM800 confocal systems /Plan apochromat 63x/1.4 objectives and Zeiss Zen 2010b version service pack1, or Nipkow Okagawa Nikon (Tokyo, Japan) spinning disk confocal imaging system Plan Apo 63x/1.4 DICIII objective and Visiview 4.0 software (Visitron Systems, Puchheim, Germany).

### 4.5. Drug Treatments

To induce sphingolipid depletion, MEF cells were treated with 0.5 µM myriocin (or methanol vehicle) from *Mycelia sterilia* in complete culture medium, with 2.5 µM fumonisin B1 (methanol vehicle) from *Fusarium moniliforme* in culture medium containing 10% FBS delipidated as above for 3 days, or with 20 µM fumonisin B1 in complete cell culture medium overnight. To induce sphingolipid production, MEF cells were treated with 0.5 mM BSA-conjugated palmitic acid complex (Cayman Chemical, Ann Arbor, MI, USA) for 4 h at 37 °C in delipidated FBS medium, and in regular medium with 100 µM Genz-123346 (123346, DMSO vehicle) overnight, or 200 µM C6-ceramide (d18:1/6:0) (860506, Avanti) for 2 h (ethanol vehicle). To induce DAG accumulation, cells were treated with 10 µM ionomycin (I9657) in regular medium for 10 min. To release ceramides locally at the plasma membrane, cells were treated with 0.5 U/mL sphingomyelinase from *Staphylococcus aureus* (S8633) for 30 min in DMEM medium without FBS and antibiotics. After treatment, cells were either placed on ice, scraped, and used for lipid extraction, or fixed with 4% PFA and mounted for microscopy. Alternatively, after sphingomyelinase treatment, cells were stained with a sphingomyelin probe EGFP-Lysenin [100] (purified from *E.coli*, see below) at 20 µg/mL in serum-free medium for 30 min at 37 °C, followed by 2 washes in PBS and fixation as described above.

### 4.6. Image Analysis

Image analysis was performed using ImageJ v1.52 [101] NIH (Bethesda, MD, USA) or, for artificial intelligence-assisted segmentation, QuPath v0.5.0 (University of Edinburgh, Edinburgh, UK) [102] was employed with a custom script developed by the University of Geneva’s Bioimaging Facility. Colocalization analysis was conducted using the ImageJ JACoP v2.0 plugin [103] that returned Mander’s coefficients for each region of interest (ROI). Mean fluorescence intensities were measured from background-subtracted images. Analyses determining the percentage of cells displaying plasma membrane localization were conducted on images randomized using the Filename_Randomizer.txt macro https://imagej.nih.gov/ij/macros/Filename_Randomizer.txt (accessed on 10 February 2023), by a blinded experimenter who counted the number of plasma membrane-positive cells and total cells in each image using the ImageJ Cell Counter plugin. For whole-cell fluorescence intensity analyses, confocal z-stacks of 9 frames spaced at 0.5 µm intervals were acquired and sum projections of stacks were created using the Z-Projection tool. Cell and background ROIs were drawn as above on the sum projections images to obtain mean total cellular fluorescence intensities. To eliminate differences between images acquired on LSM700 vs. LSM800 microscopes, a normalization factor was calculated from images taken on both microscopes from the same coverslip and applied to obtain normalized mean total cellular fluorescence intensities (nMFI).

### 4.7. Immunoblotting

Cells grown to 80–90% confluency on 6 cm dishes were washed three times with PBS, scraped off, pelleted, and lysed in RIPA buffer (Thermo 89900) with protease inhibitors (Halt Protease Inhibitor cocktail, 87785) for 30 min at 4 °C. Cell debris was pelleted at max speed for 10 min (4 °C), and the supernatant was used for loading into the gel. Protein concentration was quantitated using RotiQuant kit (Carl Roth, 0120.1). Lysates were diluted in the SDS Loading Buffer (Roti-Load, Carl Roth, K929.2) and heated at 95 °C for 5 min. A total of 15–30 μg protein was loaded and separated using SDS-PAGE in 4–12% SurePAGE 12-well pre-cast gels (Genscript, Nanjing, China). Proteins were transferred onto PVDF membranes using iBlot or iBlot2 system (Thermo). After transfer, membranes were blocked in 5%milk/TBS-1%Tween-20 (TBST) for 1 h (RT, agitation), followed by incubation with primary antibodies diluted in 3% milk/TBST overnight at 4 °C and 1 h incubation at RT with HRP-conjugated secondary antibodies and 3% milk/TBST with multiple washes in PBS between incubations with primary and secondary antibodies. Signals were detected using Immobilon HRP substrate (MilliporeSigma, WBKLS0500) on the ImageQuant LAS 4000 mini-imaging system (GE, Boston, MA, USA).

### 4.8. Lipid Extraction and Lipidomics

For lipidomics experiments, lipids were extracted following a modified MTBE protocol [48]. MEF cells grown to confluency in lipid-free or complete culture medium on 6 cm dishes were washed in cold PBS and scraped off in 1 mL PBS. The 900 µL and 100 µL cell suspensions were transferred to two separate tubes and the cells were pelleted at 400 g for 5 min (4 °C). The pellets were kept at −80 °C or used for protein quantitation (100 µL sample) using resuspension in RIPA buffer, using RotiQuant as above, or for lipid extraction (900 µL sample) as follows. First, pellets were resuspended in 100 µL LC-MS grade water, and then 360 µL methanol and the lipid standard mix (see Reagents section) were added and the tubes vigorously vortexed. Next, 1.2 mL MTBE was added, and the samples were incubated for 1 h at RT with shaking. Then, 200 µL of water was added to induce phase separation; samples were incubated at RT for 10 min followed by centrifugation at 1000× *g* for 10 min (at RT). The upper organic phase was transferred to 13 mm glass tube, 400 µL of the artificial upper phase (MTBE/methanol/water (10:3:1.5, *v*/*v*)) was added, the phase separation was repeated, and the collected second upper phase was combined with the first one. The total lipid extract was divided into 2 equal parts: one for sphingolipids (SL) and one for glycerophospholipids (TL). Extracts were dried in a speed vacuum concentrator. After drying, the samples were flushed with nitrogen unless used for further steps. The sphingolipid aliquot was treated with methylamine to remove phospholipids through de-acylation (Clarke method [104]). Then, 0.5 mL monomethylamine reagent (MeOH/H_2_O/n-butanol/Methylamine solution (4:3:1:5 *v*/*v*) was added to the dried lipids, tubes were sonicated for 6 min and vortexed, followed by s incubation for 1 h at 53 °C. Finally, the methylamine-treated fraction was dried (as mentioned above). The monomethylamine-treated lipids were subjected to desalting via extraction with n-butanol. In short, 300 µL water-saturated n-butanol (n-butanol/water 2:1 *v*/*v*, mixed and prepared at least 1 day before use) was added to the dried lipids. The samples were vortexed and sonicated for 6 min, and 150 µL LC-MS grade water was added. The mixture was vortexed thoroughly and centrifuged at 3200× *g* for 10 min. The upper phase was transferred to a 2 mL amber vial. The lower phase extraction was repeated as described above twice more, and the organic phases were combined and dried.

Targeted lipidomic analysis using mass spectrometry was performed according to prior protocols [105] as follows: TL and SL aliquots were resuspended either in 250 µL chloroform/methanol (LC-MS grade, 1:1 *v*/*v*) or directly in the sample solvent (chloroform/methanol/water (2:7:1 *v*/*v*) + 5 mM ammonium acetate). After sonication for 6 min (RT), lipid mixtures were transferred into wells of a 96-well Twin tech PCR plate (Eppendorf, Hamburg, Germany) using Hamilton syringes. Sphingolipid aliquots were diluted 1:4, and total lipids were diluted 1:10 in the sample solvent. The PCR plate was sealed with EasyPierce foil (Thermo, AB-1720), and the samples were injected into the TSQ Vantage Triple Stage Quadrupole mass spectrometer (Thermo) equipped with a robotic sample handler and nanoflow ion source Nanomate HD (Advion Biosciences, Ithaca, NY, USA). The collision energy was optimized for each lipid class. Lipid detection parameters are listed in Appendix A. Samples were injected at a gas pressure of 30 psi with a spray voltage of 1.2 kV and run on the mass spectrometer operated with a spray voltage of 3.5 kV for positive mode and 3.0 kV for negative mode and a capillary temperature set at 190 °C. Sphingolipid and glycerophospholipid species were identified and quantified using multiple reaction monitoring (MRM). Ceramide species were also quantified with a loss of water in the first quadrupole. Each biological replicate was measured twice (2 technical replicates, TR). Each TR consisted of 3 measurements for each transition. Each lipid species was quantified using standard curves created from internal standards. Data analysis was performed using R version 3.6.2 (R Foundation for Statistical Computing, Vienna, Austria) where lipid structures and their corresponding masses were matched to lipid classes according to LIPD MAPS -based nomenclature for lipidomic data [106]. The values were then normalized to the total lipid content of each corresponding lipid extract (mol% of the sum of total PC, PE, PI, and PS), except for samples of phagocytosing cells. In this case, because large changes in PC under the phagocytosing condition skewed the normalization, lipid concentrations were normalized first to total protein of each corresponding individual sample, then to the average of the sum of total PC, PE, PI, and PS for the entire sample group (control and phagocytosing) processed the same day. Lipid values are thus expressed as mol% of the daily average total lipid mass (mol% of ave tot). Values greater than 100% thus represent values greater than the average total glycerophospholipid.

### 4.9. Protein Purification from E. coli

Rosetta 2(DE3) *E. coli* cells (Novagen, Pretoria, South Africa) were transformed with the plasmids expressing His6-SUMO3-tagged KSR probes or GFP alone (control): pETM11-SUMO3-sfGFP-KSR1-CA3 (N-KSR)-, pETM11-SUMO3-KSR1-CA3-sfGFP (C-KSR), and pETM11-His6-SUMO3-sfGFP. Transformed bacteria were grown at 37 °C in 1 L LB medium containing 100 µg/mL kanamycin and 34 µg/mL chloramphenicol until OD600 0.6–0.8. Protein expression was induced through the overnight incubation with 0.4 mM isopropyl β-D-thiogalactoside (IPTG) at 16 °C. The next day bacteria were lysed using the Microfluidizer LM20 high-pressure homogenizer (15,000 psi pressure, 2 cycles) in lysis buffer (50 mM Tris pH 7.5, 500 mM NaCl, 20 mM imidazole, protease inhibitors (1 tablet/50 mL), 0.5 mM DTT) and the soluble fraction was obtained using centrifugation at 16,000× *g* for 1 h (4 °C). Proteins harboring the His6 tag were captured using incubation with Ni-NTA resin (Qiagen, Venlo, Netherlands) (2 mL resin per bacterial lysate from 1 L culture) for 2 h at 4 °C and 30 rpm rotation. Agarose resin with bound proteins was centrifuged at 1200 rpm for 8 min (4 °C), and the slurry was transferred to a column with a cotton filter. The column was washed with 20 mL lysis buffer, His6-tagged proteins were eluted with 6 mL Elution Buffer (50 mM Tris pH 7.5, 250 mM NaCl, 500 mM imidazole), DTT was added to 5 mM to prevent protein aggregation. Eluted proteins were dialyzed against 50 mM Tris pH 7.5, 200 mM NaCl, 1 mM DTT overnight at 4 °C. The His6-SUMO3 tag was cleaved off the proteins using incubation with His-tagged SenP2 protease (EMBL Protein Expression and Purification Core Facility European, Heidelberg, Germany) at a molar ratio of SenP2 to protein 40:1. The undigested proteins and the protease were removed using Ni-NTA resin, purified proteins were eluted by applying Elution Buffer 2 (50 mM Tris pH 7.5, 250 mM NaCl, 10 mM imidazole) to Ni-NTA-packed column. The quality of purification was assessed using SDS-PAGE followed by Coomassie staining (RotiBlue kit, A152.1, CarlRoth). sfGFP-N-KSR was further purified using size exclusion chromatography on the S75 column pre-equilibrated with 50 mM Tris at pH 7.4 and 250 mM NaCl.

### 4.10. Cell Lysate Preparation from HEK Cells

HEK cells grown on 10 cm dishes in HEK complete culture medium to 70–80% confluency were transfected as above with plasmids encoding EGFP (control), KSR lipid probes, or PKC-C1(2)-GFP, where 20 µg plasmid DNA/50 µL lipofectamine 2000 was used per dish. After overnight incubation, the cells were trypsinized for 1 min (RT) and resuspended in PBS containing 0.5 mM EDTA. Cells suspension was centrifuged 200× *g* for 5 min (4 °C), followed by a wash in PBS. After pelleting, cells were resuspended in 120 µL hypotonic lysis buffer (10 mM Tris-HCl, pH = 7.5, Halt protease and phosphatase inhibitors) and incubated on ice for 20 min. Cells were lysed by passing the suspension 15 times through a BD Microlance 30 G syringe needle (BD, Eysins, Switzerland, 304000). Protein concentration was measured using a NanoDrop (Thermo). Finally, NaCl and DTT were added to 150 mM and 5 mM, respectively.

### 4.11. Liposome Microarray-Based Assay (LiMA)

Liposome-based protein–lipid interaction assay was performed following the protocol described in [61]. Lipid mixtures of different compositions were prepared in chloroform/methanol/water (20:9:1 *v*/*v*/*v*). DOPC-based liposomes contained 0.1% mol fluorescent lipid PE-Atto647 (Atto Tec, Siegen, Germany); 0.5% mol stabilizing lipid PE-PEG350; and 2%, 5%, or 10% of bovine ceramide mixture (CerMix), C18-ceramide (C18Cer), C18-dihydroceramide (DHCer), sphingomyelin (SM), glucosyl-ceramide (GlcCer), sphingosine (Sph), sphingosine-1-phosphate (S1P), ceramide-1-phosphate (Cer1P), and diacylglycerol (DAG); or 10% phosphatidylinositol-4,5-bisphosphate (PI(4,5)P2) (see also Appendix A). IPM-mimicking liposomes were based on a POPC carrier lipid backbone with 0.1% mol fluorescent lipid PE-Atto647; 0.5% mol stabilizing lipid PE-PEG350; 33% cholesterol; 21.2% DOPS; 15% POPE; and 2%, 5%, or 10% CerMix, C18DHCer, C18Cer, DAG, POPA, SM, and Sph. Mixtures containing 5% and 10% GlcCer or 10% DAG did not form liposomes, and all liposomes containing DOPA were autofluorescent; thus, these spots were excluded from the analysis. Lipids were spotted onto thin agarose layers covered by the protective membrane on top of a glass slide using automated lipid spotting under an inert argon atmosphere (Automatic TLC-spotter4, Camag, Muttenz, Switzerland). A microfluidic chamber with four channels was manufactured from polydimethylsiloxane (PDMS), as described in [61]. Then, the agarose layers with spotted lipids were glued onto PDMS chambers using a 2:1 mixture of silicone elastomer/curing agent diluted 3× (*v*/*v*) in hexane as an adhesive, so the resulting microfluidic chamber contained four channels of 30 spots, with 120 lipid spots per chamber. Liposomes were generated by manually injecting the liposome buffer (10 mM HEPES, 150 mM NaCl) into each channel with a Hamilton syringe. Then, 40 µL of the solution containing purified proteins at 5–10 µM or HEK cell lysates in liposome buffer with 5 mM DTT was injected into channels using an automated injection system with a flow rate of 8 µL/min. After incubation of freshly formed liposomes with the GFP-tagged proteins for 1 h at RT, the channels were manually washed with 80 µL of liposome buffer, and all spots in the chamber were imaged with a Zeiss Axio Observer Z1 video time-lapse system. All images were taken with the 20× objective; liposomes were imaged with the Cy5 filter set (5 and 10 ms exposure), and GFP-tagged proteins were imaged at different exposure times (1, 5, 10, 25, 50, 75, 100, 150, 200, and 300 ms).

Images were analyzed using CellProfiler version 4.2.5 software (Broad Institute, Cambridge, MA, USA) using a custom pipeline described in detail in [61]. A custom-made R script was used to calculate the normalized binding intensity value (NBI) for each protein–lipid interaction. NBI represents a mean of the ratio of GFP to Atto647 fluorescence divided by exposure time at all exposure times for which spots were successfully segmented, and is proportional to the amount of GFP-tagged interaction protein recruited to the liposome surface. NBI scores were then expressed as the Log2 ratio of liposome of interest to control liposome-containing carrier, fluorescent, and stabilizer lipids only.

### 4.12. Preparation of Phagocytic Beads and Phagocytosis

The 3 μm carboxyl polystyrene beads (Spherotech, Lake Forest, IL, USA, CP30-10) were covalently coupled with purified human IgG (hIgG, Dunn Labortechnik, Thelenberg, Germany) by washing 3 times in sterile PBS with centrifugation at maximum speed (18,000× *g*) at 4 °C, activating them with 50 mM 1-ethyl-3-(3-dimethylaminopropyl) carbodiimide hydrochloride (EDC-HCl, Carl Roth, 2156.1) for 15 min in PBS at room temperature (RT) with rigorous shaking, followed by 3 washes at 4 °C in 0.1 M Na_2_B_4_O_7_ buffer (pH 8.0). Finally, 2 µg/µL of hIgG (and 15 ng/µL iFluor647-amine (AAT Bioquest, Pleasanton, CA, USA), for fluorescent beads) was added to the beads and incubated overnight at 4 °C on a shaker. The next day, the beads were washed twice with 250 mM glycine/PBS followed by two washes in PBS. To render MEFs phagocytic, cells were transfected with 1 µg/2 × 10^5^ cells pcDNA3-myc-FCGR2A (Fc receptor) plus KSR1-based probes and, where indicated, TagRFP-KDEL, mCherry-STIM1, or mCherry-Sec22b at 0.3 µg per 2 × 10^5^ cells. Cells were washed in complete medium 4–6 h after transfection. The following day, beads were added to the cells at a bead-to-cell ratio 10:1, cells were centrifuged at 200× *g* for 2 min at RT, incubated with the beads for the indicated times at 37 °C, washed with PBS, and fixed with 4% PFA/PBS for 30 min, followed by three washes with PBS and mounting as described above.

### 4.13. Statistics and Reproducibility

All statistical analyses were performed using Prism v10.2.0 (GraphPad Software, La Jolla, CA, USA). P values that are significant are shown above the bars. Two-tailed unpaired *t*-test were performed when comparing two samples, Welch’s correction was used if F test showed a significant difference in variance. For three or more sample comparisons, a one-way ANOVA was employed in case of negative F test, and Brown–Forsythe and Welch’s test were performed in case F test showed significant difference in variance, with Dunnett’s post-hoc test. For lipidomic data, multiple ratio-paired t-test was performed with a two-stage step-up method (Benjamini, Krieger, and Yekutieli), with a false discovery rate set at 5%. Categorical quantification of plasma membrane recruitment was performed using a blinded experimenter on randomized images. Contingency tables were analyzed using a Chi-squared test. Cells exhibiting abnormal morphology (multinucleated, high vacuolation) or signs of death (blebbing) were excluded from analysis. A minimum of 5 replicate fields per coverslip were imaged for analysis of fixed cells. These may contain from one to tens of cells or one to tens of phagosomes which are considered as replicate values. For liposome analysis, spots with large aggregates were excluded from the analysis.

## Figures and Tables

**Figure 1 ijms-25-02996-f001:**
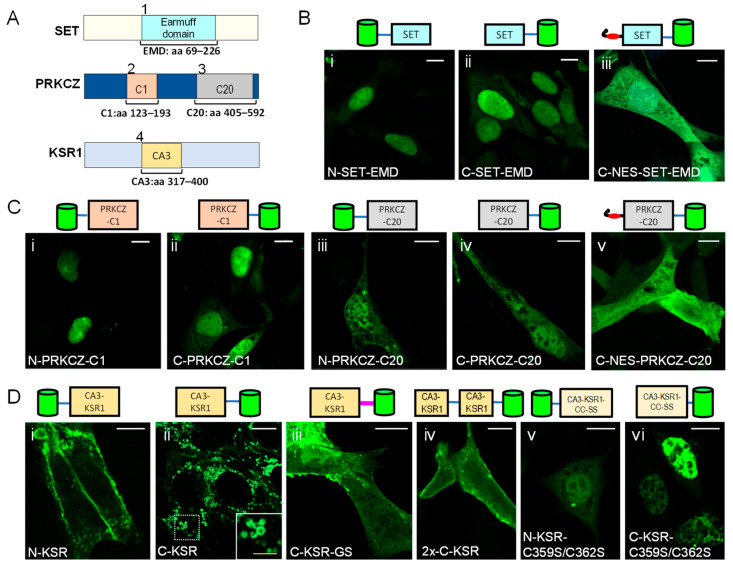
Localization of candidate ceramide probes in mouse embryonic fibroblasts (MEFs). (**A**) Schematic representation of ceramide-binding domains used for probe construction from the following source proteins: Human SET earmuff domain (EMD, amino acids (aa) 69–226); atypical PKC type zeta C1 domain (PRKCZ-C1, aa 123–193), and C20 domain (PRKCZ-C20, aa 405–592); and KSR1 CA3 domain (KSR, aa 317–400). (**B**–**D**) Confocal images of MEF cells transfected with N-terminally (N-) or C-terminally (C-) EGFP-tagged ceramide-binding domains of SET (**B**), PRKCZ (**C**), and KSR1 (**D**). Constructs preceded by a nuclear export signal are labelled NES, the construct bearing a glycine–serine linker is GS, the tandem construct is 2x, and the cysteine mutants are C359S/C362S. The inset in C-KSR highlights enlarged vesicular structures. White bars = 10 µm, yellow bar = 3 µm.

**Figure 2 ijms-25-02996-f002:**
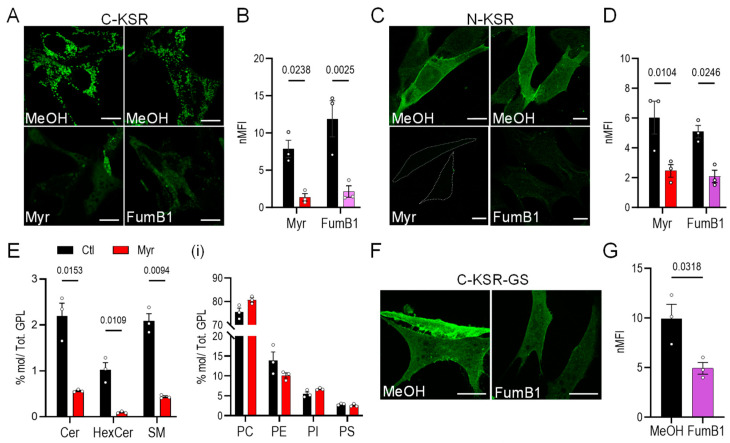
Response of KSR probes to sphingolipid depletion. (**A**–**D**) Confocal images of MEF cells stably expressing C-KSR (**A**) or N-KSR (**C**) (in green) treated with 0.5 µM myriocin (Myr, **left**) or 2.5 µM fumonisin B1 (FumB1, **right**) for 3 days, and respective methanol (MeOH) vehicle controls. The quantification of the normalized total cellular mean fluorescence intensity (nMFI) of cells in (**A**) and (**C**) are shown in (**B**) and (**D**), respectively. (*n* = 3 coverslips for each condition). White bars = 10 µm. (**E**) Targeted lipidomic analysis of MEF cells stably expressing C-KSR, treated with Myr as above. Left panel shows the quantification of the sum total mass of lipids classified as ceramides (Cer), hexosyl-ceramides (HexCer), and sphingomyelins (SM), normalized to the total mass glycerophospholipids (GPL) in the same sample (expressed as percent mole, %mol/Tot. GPL). Inset (**i**) shows the quantification of total major phospholipid species phosphatidylcholine (PC), phosphatidylethanolamine (PE), phosphoinositide (PI), phosphatidylserine (PS). (*n* = 3 independent biological samples, see also Appendix A). (**F**,**G**) Similar analysis as in (**A**–**D**) of cells transiently transfected with C-KSR-GS (green) and treated overnight with 10 µM FumB1. Control and treated images are taken using identical imaging settings and shown using identical contrast settings. Bars are means +/− SEM. (*n* = 3 coverslips).

**Figure 3 ijms-25-02996-f003:**
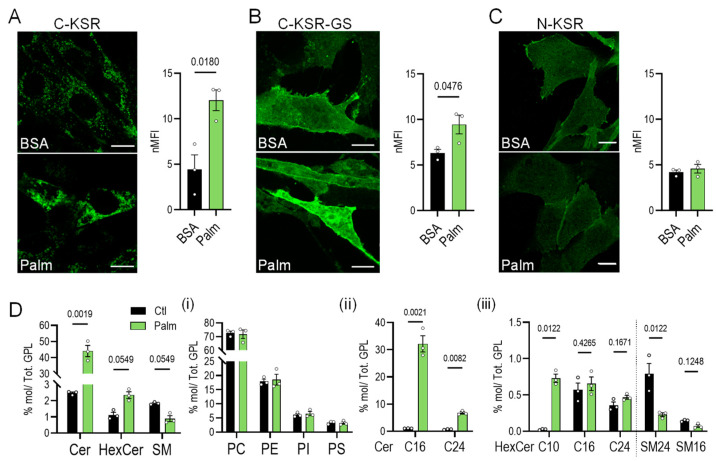
Response of KSR probes to palmitate-induced ceramide accumulation. (**A**–**C**) Confocal images of MEF cells expressing C-KSR (**A**), C-KSR-GS (**B**), and N-KSR (**C**) (in green) treated with 0.5 mM BSA-palmitate (Palm) or BSA control for 4 h. The quantification of the total cellular fluorescence (nMFI) is shown to the right of the respective images (*n* = 3 coverslips for each condition). (**D**) Targeted lipidomic quantification of total Cer, HexCer, and SM in MEF cells expressing C-KSR, treated with Palm as above. Inset (**i**) shows the quantification of PC, PE, PI, and PS. Inset (**ii**) shows quantification of C16- and C24-Cer. Inset (**iii**) shows quantification of C10-, C16-, and C24-HexCer, as well as C24 and C16-SM (labelled SM24 and SM16, respectively) (*n* = 3 independent biological samples, see also Appendix A). Control and treated images were taken using identical imaging settings and shown using identical contrast settings. Bars are means +/− SEM. (*n* = 3 coverslips). White bars = 10 µm.

**Figure 4 ijms-25-02996-f004:**
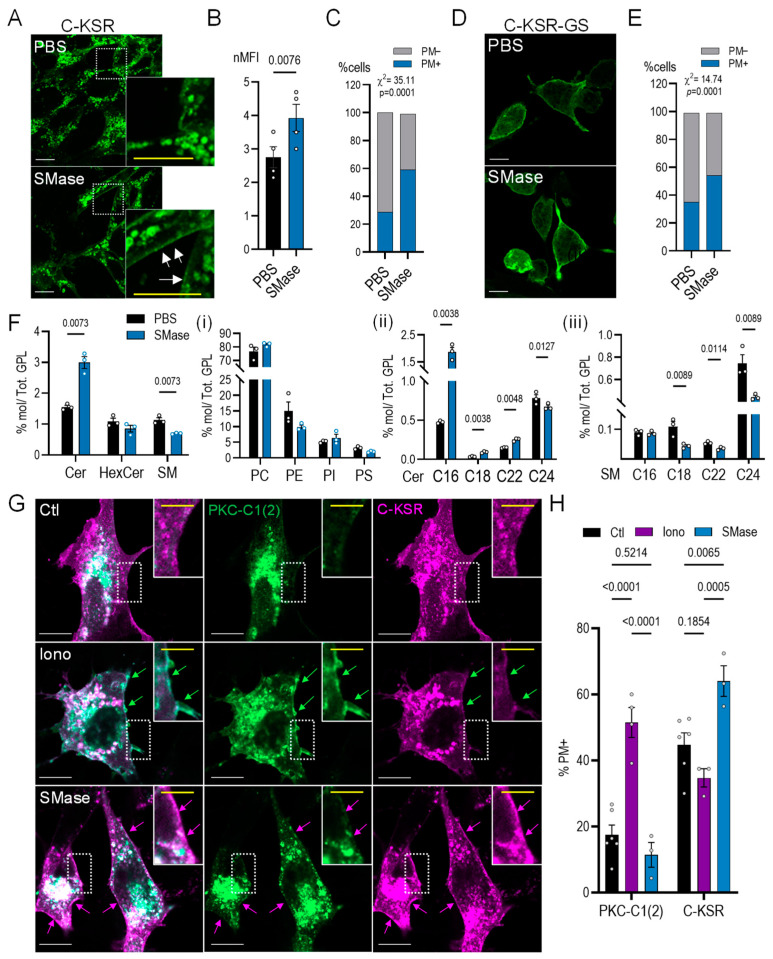
Response of KSR probes to sphingomyelinase-induced ceramide accumulation. (**A**–**E**) Confocal images and analysis of C-KSR (**A**–**C**) and C-KSR-GS (**D**,**E**) in cells treated for 30 min with 0.5 U/mL sphingomyelinase (SMase). In (**B**) a quantification of total cellular fluorescence (nMFI) is shown, whereas (**C**) and (**E**) show quantification of the fraction of cells displaying plasma membrane localization (indicated by white arrows in (**A**) insets) of C-KSR and C-KSR-GS, respectively (*n* = 4 coverslips). (**F**) Targeted lipidomic quantification of total Cer, HexCer, and SM in MEF cells expressing C-KSR, treated with SMase as above. Inset (**i**) shows the quantification of PC, PE, PI, and PS. Inset (**ii**) shows quantification of C16-, C18-, C22-, and C24-Cer. Inset (**iii**) shows quantification of C16-, C18-, C22-, and C24-SM (*n* = 3 independent biological samples, see also Appendix A). (**G**,**H**) Confocal images of MEF cells co-transfected with C-KSR-mRFP (magenta) and diacylglycerol (DAG) probe PKC-C1(2)-EGFP (green), control (Ctl, **top**) or treated with 10 µM ionomycin (Iono) for 10 min (**middle**), or with SMase as above (**bottom**). Green arrows indicate regions of PM localization of PKC-C1(2). Magenta arrows indicate regions of PM localization of C-KSR-mRFP. (**H**) Quantification of cells in (**G**) displaying PM localization of either C-KSR or PKC-C1(2) probe (*n* = 6/3/3 coverslips for Ctl/Iono/SMase). Control and treated images were taken using identical imaging settings and shown using identical contrast settings. Bars are means +/− SEM. White bars = 10 µm, yellow bar = 3 µm.

**Figure 5 ijms-25-02996-f005:**
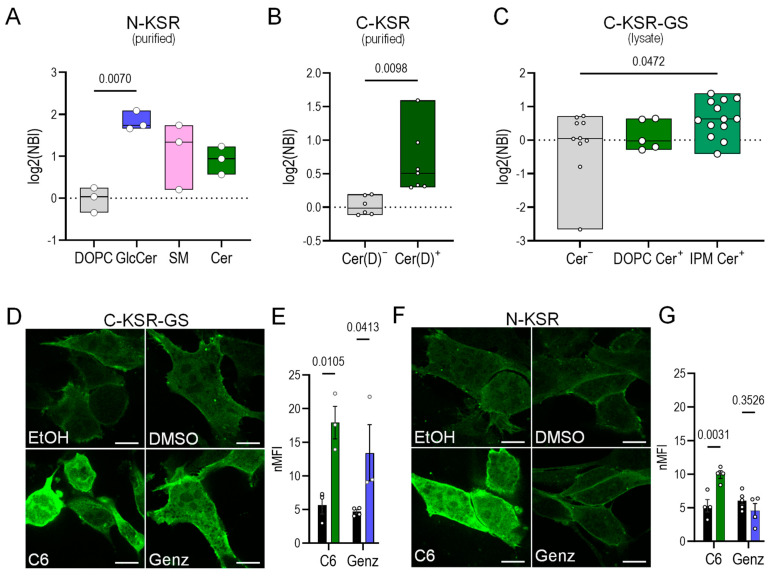
Differential lipid preference of C- and N-tagged KSR probes. (**A**) Liposome microarray (LiMA) analysis using purified N-KSR. Binding scores of control liposomes containing dioleoyl-phosphatidylcholine (DOPC), and DOPC liposomes containing 2% glucosyl-ceramide (GlcCer), 10% sphingomyelin (SM), or 10% ceramide mixture (Cer) are shown. Binding scores are expressed as log2-transformed normalized binding intensity (NBI) score ratios to control (*n* = 3 independent biological samples). (**B**) LiMA analysis using purified C-KSR. Pooled binding scores of liposomes either lacking (Cer(D)^−^) or containing ceramide or ceramide derivatives (Cer(D)^+^). Conditions included were as follows: Cer(D)^−^: DOPC alone, and 10%PI(4,5)P2 or 10% sphingosine in DOPC. Cer(D)^+^: 10% ceramide mix (Cer), SM, GlcCer, or Cer1P in DOPC (*n* = 6/7 Cer(D)^−^/Cer(D)^+^ independent biological samples). (**C**) LiMA analysis of C-KSR-GS lysates pooled binding scores of control liposomes (Cer^−^) or containing 5% and 10% ceramides in either a DOPC or inner plasma membrane (IPM) mimicking composition (DOPC Cer^+^ and IPM Cer^+^, respectively). Conditions included were as follows: Cer^−^: DOPC, POPC, and IPM control liposomes; DOPC Cer^+^: C18-ceramide in DOPC; IPM Cer^+^: ceramide mix and C18-ceramide (*n* = 10/5/13 Cer^−^/ DOPC Cer^+^/IPM Cer^+^ independent biological samples). (**D**–**G**) Confocal images of MEFs expressing C-KSR-GS (**D**) or N-KSR (**F**) and treated (**bottom**) with either Cer analog C6-ceramide (200 µM, 2 h, left) or 100 µM of the GlcCer synthase inhibitor Genz-123346 (Genz, 1d, **right**), or their respective vehicle controls ethanol (EtOH) or dimethylsulfoxide (DMSO) (**top**). (**E**,**G**) show the quantification of total cellular fluorescence (nMFI) of C-KSR-GS and N-KSR, respectively (in (**E**): *n* = 3/3/4/3 coverslips for EtOH/C6/DMSO/Genz; in (**G**): *n* = 4 coverslips for all conditions). Control and treated images were taken using identical imaging settings and shown using identical contrast settings. In (**A**–**C**) the box plots show the range (min to max) and the line shows the median value. In (**E**,**G**) the bars are means +/− SEM. White bars = 10 µm.

**Figure 6 ijms-25-02996-f006:**
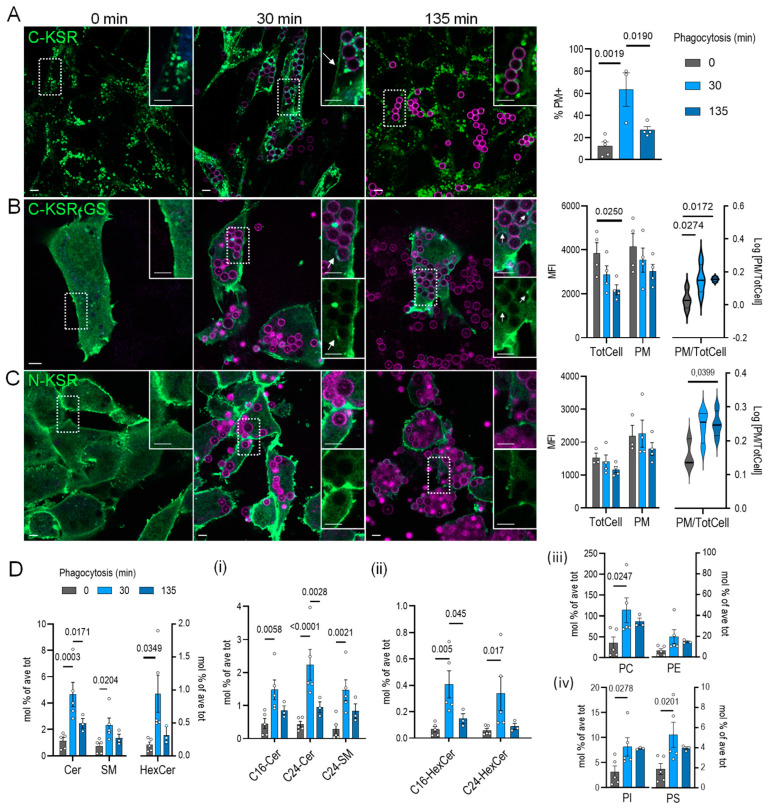
Behavior of KSR probes during phagocytosis in MEFs. (**A**–**C**) Confocal images of phagocytic MEFs expressing (in green) C-KSR (**A**), C-KSR-GS (**B**), and N-KSR (**C**) exposed to IgG-AlexaFluor647-coupled beads at 1:10 cell/bead ratio for 0 (**left**), 30 (**middle**), and 135 (**right**) min. Percentage of C-KSR plasma membrane recruitment (%PM+) during phagocytosis is shown to the right of the corresponding images (*n* = 5/3/4 coverslips for Ctr/30 min/135 min). MFI for total cell (TotCell) and PM are shown to the right of corresponding images for C-KSR-GS and N-KSR (*n* = 4 coverslips for all conditions with C-KSR-GS, and *n* = 3/4/4 coverslips for Ctr/30 min/135 min with N-KSR). The white arrows in (**A**) indicate PM recruitment of C-KSR, whereas the white arrows in (**B**) indicate peri-phagosomal accumulation of C-KSR-GS. The corresponding violin plots on the right of the bar graphs show the range (min to max) and distribution, with a line showing the median value of the logarithm of the average PM/TotCell ratio per coverslip of the same data as the bar graphs. (**D**) Targeted lipidomic quantification of MEF cells stably expressing C-KSR, transfected with FGCR2A-c-myc and exposed to IgG-beads as above. The left panel shows total Cer, HexCer, and SM. Inset (**i**) shows quantification of C16-Cer, C24-Cer, and C24-SM. Inset (**ii**) shows quantification of C16-HexCer and C24-HexCer. Insets (**iii**) and (**iv**) show quantification of total phospholipids PC and PE in (**iii**), and PI and PS in (**iv**) (*n* = 3 independent biological samples, see also Appendix A). Bar graphs are means +/− SEM. White bars = 10 µm.

**Figure 7 ijms-25-02996-f007:**
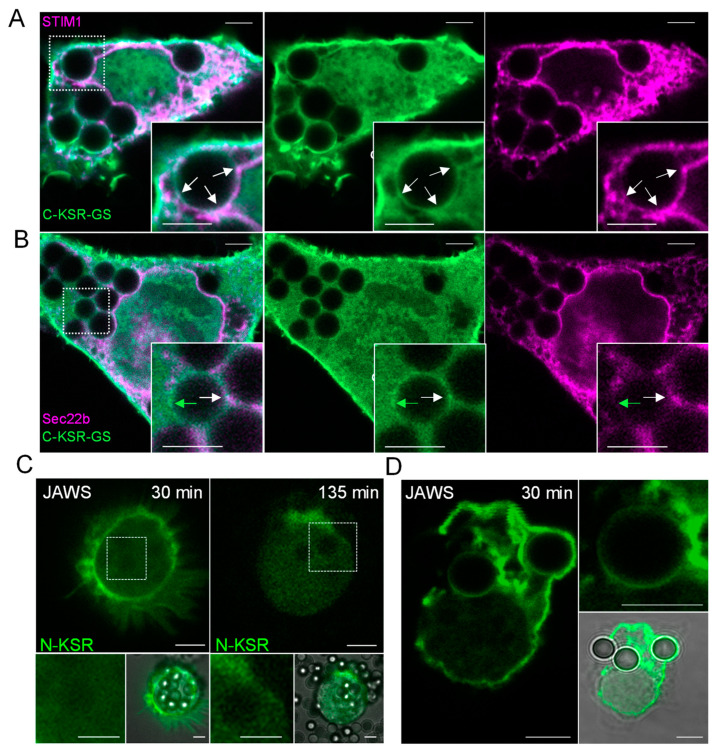
Behavior of KSR probes during phagocytosis upon ER protein overexpression and in dendritic cells. (**A**,**B**) Confocal images of phagocytic MEFs co-transfected with C-KSR-GS (green) and membrane contact site proteins mCh-STIM1 ((**A**), magenta) and mCh-Sec22b ((**B**), magenta), and exposed to IgG-beads for 30 min (see also Figure 6). White arrows show periphagosomal regions where overlap of fluorescent protein signal is discerned, green arrow shows regions with green fluorescence only. (**C**,**D**) Confocal images of dendritic cell line JAWS transfected with (in green) N-KSR (**C**) exposed to IgG-coupled beads at 1:10 cell/bead ratio for 30 (**left**) and 135 (**right**) min, and C-KSR-GS (**D**) exposed to beads for 30 min. Brightfield images are shown to confirm presence of internalized beads. Images in (**A**,**B**) and (**C**,**D**) were taken using identical imaging settings and shown using identical contrast settings. Bars = 3 µm.

**Figure 8 ijms-25-02996-f008:**
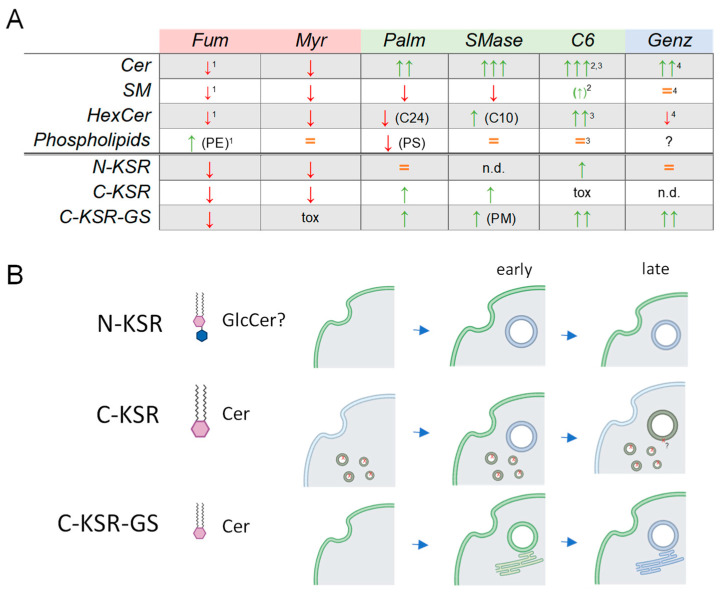
Summary and model. (**A**) Table summarizing effect of ceramide-modifying treatments used in this study on major lipid classes (ceramides (Cer), sphingomyelins (SM), hexosyl-ceramides (HexCer), and phospholipids) and on the three KSR1-based probes. In cases where only certain sub-species of lipids were affected, the major sub-species affected are listed in brackets. Decreases are shown as a red arrow pointing down, increases as green arrow(s) pointing up, and similar levels as an orange equals sign. Tox = toxicity. n.d. = not determined. PM = plasma membrane. Myr (myriocin), Palm (palmitate), and SMase (sphingomyelinase) effects were measured in this study. The effects of Fum (fumonisin B1), C6 (C6-ceramide), and Genz (Genz-123346) were extrapolated from the literature. (^1^ [50]; ^2^ [66]; ^3^ [67]; ^4^ [68]). (**B**) Hypothetical model of the behavior of N-KSR, C-KSR, and C-KSR-GS before and during phagocytosis progression (blue arrows). N-KSR (**top**), with a putative sensitivity for glucosyl-ceramides (GlcCer) is relatively enriched at the PM in later time phagocytic time points. C-KSR (**middle**) and C-KSR-GS (**bottom**) are both more sensitive to ceramides (Cer). Higher dynamic range and turnover of C-KSR may help more sensitive detection of ceramide changes at the PM during early stages of phagocytosis, but presence in lysosomes and phagosomes may represent a partially degraded probe. The more homogeneous and higher cytosolic expression of C-KSR-GS may make it most suitable for detecting ceramide changes on the cytosolic leaflet of organellar membranes such as the phagosome and endoplasmic reticulum. Created with BioRender.com.

## Data Availability

Full datasets, including all microscopy and Western blot source images, are made available on the University of Geneva’s FAIR-compliant data repository Yareta (DOI: 10.26037/yareta:nl3snh47anf2rnfvd3o4dir5ju) under a CC BY 4.0 license. Plasmids generated in this study are made available via Addgene upon publication, to be distributed subject to a standard Material’s Transfer Agreement.

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
