# Peer review of "Development of Genetically Encoded Fluorescent KSR1-Based Probes to Track Ceramides during Phagocytosis"

_ijms, 2024, doi:10.3390/ijms25052996_

Round 1
Reviewer 1 Report
Comments and Suggestions for Authors
The comments in the attachment

English is fine
Author Response
We would like to thank all of the reviewers for their time and effort in providing us with constructive comments that we believe have helped us to make our study stronger. Please find below the reviewer comments and our responses in italics.
Comment1: The manuscript "Development of genetically-encoded fluorescent KSR1-based probes to track ceramides during phagocytosis" focuses on the creation of novel genetically encoded probes for ceramide imaging.
The manuscript describes the construction of the first genetically encoded probes for visualization of ceramides. Principles of these constructs are well-known for other purposes, so the approach is reasonable. Although there are other methods for quantification of ceramides, including lipidomics, applied in this work to control the results, rapid visualization of sphingolipids in living cells is a fruitful contribution in the field.
The work involves constructing 14 original biosensors using ceramide-binding protein domains reported in the literature. The study extensively examines the most promising sensors based on the KSR1 CA3 domain, revealing differences in their localization and binding to ceramide analogues. These probes are then applied to study phagocytosis. The manuscript is well-written, with adequately described methods, and the conclusions are well-supported by the results. The constructed probes could be valuable for further lipid studies in living cells.
The methodological part of the study quite solid, the constructed probes were tested using various approaches. Accumulation of the target molecules for imaging was corroborated by other approaches. The results support the selection of the targeting moiety, differences in specificity depending of binding site studied in detail. The discussion is supported by relevant references, demonstrating the applicability of the constructed probes for molecular biology.
Response1: We thank the reviewer for these positive comments.
Comment2: The manuscript is suitable for publication in IJMS, with only minor comments:
1. The figures in the manuscript have poor quality.
Response2: High quality images are now provided.
Comment3: There are some typos and formatting errors in the text (some are marked in the attached PDF). Careful proofreading is advised.
Response3: We apologize for these mistakes many introduced during the formatting conversion. We have carefully gone over the manuscript and corrected errors.

Reviewer 2 Report
Comments and Suggestions for Authors
The manuscript "Development of genetically-encoded fluorescent KSR1-based probes to track ceramides during phagocytosis" focuses on the creation of novel genetically encoded probes for ceramide imaging.
The manuscript describes the construction of the first genetically encoded probes for visualization of ceramides. Principles of these constructs are well-known for other purposes, so the approach is reasonable. Although there are other methods for quantification of ceramides, including lipidomics, applied in this work to control the results, rapid visualization of sphingolipids in living cells is a fruitful contribution in the field.
The work involves constructing 14 original biosensors using ceramide-binding protein domains reported in the literature. The study extensively examines the most promising sensors based on the KSR1 CA3 domain, revealing differences in their localization and binding to ceramide analogues. These probes are then applied to study phagocytosis. The manuscript is well-written, with adequately described methods, and the conclusions are well-supported by the results. The constructed probes could be valuable for further lipid studies in living cells.
The methodological part of the study quite solid, the constructed probes were tested using various approaches. Accumulation of the target molecules for imaging was corroborated by other approaches. The results support the selection of the targeting moiety, differences in specificity depending of binding site studied in detail. The discussion is supported by relevant references, demonstrating the applicability of the constructed probes for molecular biology.
The manuscript is suitable for publication in IJMS, with only minor comments:
1. The figures in the manuscript have poor quality.
2. There are some typos and formatting errors in the text (some are marked in the attached PDF). Careful proofreading is advised.

Author Response
We would like to thank all of the reviewers for their time and effort in providing us with constructive comments that we believe have helped us to make our study stronger. Please find below the reviewer comments and our responses in italics.
Comment1: The manuscript "Development of genetically-encoded fluorescent KSR1-based probes to track ceramides during phagocytosis" focuses on the creation of novel genetically encoded probes for ceramide imaging.
The manuscript describes the construction of the first genetically encoded probes for visualization of ceramides. Principles of these constructs are well-known for other purposes, so the approach is reasonable. Although there are other methods for quantification of ceramides, including lipidomics, applied in this work to control the results, rapid visualization of sphingolipids in living cells is a fruitful contribution in the field.
The work involves constructing 14 original biosensors using ceramide-binding protein domains reported in the literature. The study extensively examines the most promising sensors based on the KSR1 CA3 domain, revealing differences in their localization and binding to ceramide analogues. These probes are then applied to study phagocytosis. The manuscript is well-written, with adequately described methods, and the conclusions are well-supported by the results. The constructed probes could be valuable for further lipid studies in living cells.
The methodological part of the study quite solid, the constructed probes were tested using various approaches. Accumulation of the target molecules for imaging was corroborated by other approaches. The results support the selection of the targeting moiety, differences in specificity depending of binding site studied in detail. The discussion is supported by relevant references, demonstrating the applicability of the constructed probes for molecular biology.
Response1: We thank the reviewer for these positive comments.
Comment2: The manuscript is suitable for publication in IJMS, with only minor comments:
1. The figures in the manuscript have poor quality.
Response2: High quality images are now provided.
Comment3: 2. There are some typos and formatting errors in the text (some are marked in the attached PDF). Careful proofreading is advised.
Response3: We apologize for these mistakes many introduced during the formatting conversion. We have carefully gone over the manuscript and corrected errors.

Reviewer 3 Report
Comments and Suggestions for Authors
The study focuses on the development of genetically-encoded fluorescent probes based on the Kinase Suppressor of Ras 1 (KSR1) to track ceramide dynamics during phagocytosis. Fourteen Enhanced Green Fluorescent Protein (EGFP) fusion constructs were generated and screened. The key findings include: Three probes (N-KSR, C-KSR, C-KSR-GS) showed differential responses to ceramide levels and localization in cells. The probes responded distinctly to treatments modifying ceramide levels, suggesting varying specificities for ceramide and possibly its derivatives. During phagocytosis, the probes demonstrated unique dynamics, indicating their potential for analyzing sphingolipid dynamics in living cells. Overall, the manuscript presents significant advancements in the development of ceramide probes, with potential implications for understanding lipid dynamics in cellular processes.
Major Concerns:
1. There is a lack of discussion on the potential safety concerns and clinical implications of using these probes in therapeutic contexts.
2. The manuscript could benefit from a more in-depth exploration of the underlying mechanisms driving the differential responses of the probes.
3. The applicability of these findings to other cellular processes beyond phagocytosis needs to be addressed.Specificity and Selectivity of Probes: The manuscript might need a more detailed evaluation of the specificity and selectivity of the probes for ceramide versus other sphingolipids or cellular components, which is crucial for their effective application.
Minor Concerns:
1. Concerns regarding the stability of the probes under different experimental conditions and their resistance to photobleaching, which is essential for their utility in long-term imaging studies.
2. Potential interference of the probes with normal cellular processes, including whether their presence affects the natural dynamics of ceramide or cellular responses during phagocytosis.
3. The study could benefit from a more robust quantitative analysis of the fluorescence changes, including calibration methods to correlate fluorescence intensity with ceramide concentrations.
4. A comparative analysis with existing methods for tracking ceramide dynamics, to highlight the advantages or limitations of the proposed probes.
5. Discussion of potential artifacts in the imaging process and limitations of the probes, including their potential to aggregate or mislocalize within the cell.
Addressing these concerns would provide a more comprehensive evaluation of the probes' capabilities and limitations, enhancing the manuscript's overall scientific rigor and impact.
Comments on the Quality of English LanguagePlease consider polishing your writing to improve flow and readability.
Author Response
We would like to thank all of the reviewers for their time and effort in providing us with constructive comments that we believe have helped us to make our study stronger. Please find below the reviewer comments and our responses in italics.
Comment1: The study focuses on the development of genetically-encoded fluorescent probes based on the Kinase Suppressor of Ras 1 (KSR1) to track ceramide dynamics during phagocytosis. Fourteen Enhanced Green Fluorescent Protein (EGFP) fusion constructs were generated and screened. The key findings include: Three probes (N-KSR, C-KSR, C-KSR-GS) showed differential responses to ceramide levels and localization in cells. The probes responded distinctly to treatments modifying ceramide levels, suggesting varying specificities for ceramide and possibly its derivatives. During phagocytosis, the probes demonstrated unique dynamics, indicating their potential for analyzing sphingolipid dynamics in living cells. Overall, the manuscript presents significant advancements in the development of ceramide probes, with potential implications for understanding lipid dynamics in cellular processes.
Major Concerns:
- There is a lack of discussion on the potential safety concerns and clinical implications of using these probes in therapeutic contexts.
Response1: We agree with the reviewer that since KSR1 is an oncogene, readers should be aware of any potential safety issues in working with our probes. We now briefly discuss (Pg 14 Lines 403-418) the role KSR1 as a regulator of the RAS-RAF-MEK-ERK signaling pathway which is well known for its role in oncogenesis. We also point out that the CA3 ceramide-binding domain we employ is only a small portion (83 out of 923 amino acids in total) of the much larger protein KSR1, composed of 5 conserved domains CA1-CA5, where the ERK and MEF2 interacting domains (CA4-CA5), missing in our constructs, have been shown to be the ones essential in regulating downstream signaling. Thus, our constructs are not expected to be oncogenic, though following biosafety procedures is always recommended. Indeed, the difficulty in establishing the C-KSR-GS stable cell line, where cells grew very slowly or died, phenocopies the effect of KSR1 knock-down and might rather act in a dominant-negative fashion. In terms of clinical implications, we now mention (Pg 17 Lines 518-524) that ceramide dysregulation has been linked to several pathologies including metabolic, cardiovascular and neurodegenerative diseases. Also, since ceramide binding is one of the early steps in KSR1 activation, the constructs in this study or future modifications thereof maybe useful not only to identify novel drugs that impair KSR1 function but also to identify new drugs modifying ceramide metabolism that may be useful to understand and treat such diseases.
Comment2: The manuscript could benefit from a more in-depth exploration of the underlying mechanisms driving the differential responses of the probes.
Response2: We have now added a more in-depth discussion of potential mechanisms underlying the differential response of the N- and C-KSR probes in Pg 15 lines 460-480. In brief, we now discuss reasons why N-terminal vs C-terminal tagging might change the specificity of the KSR1 CA3 domain, which we hypothesize maybe related to the fact that the ceramide binding pocket is quite shallow and thus may be easily deformed by the N-terminal tagging. A wider pocket might decrease the original affinity for ceramide itself and instead now favor binding ceramide species bearing a small additional head-group such as a single sugar or phosphate group. To verify such a hypothesis, however, would require a comparison of the structures of N-KSR and C-KSR-GS, a complex undertaking that would be extremely interesting to pursue in future studies.
Comment3: The applicability of these findings to other cellular processes beyond phagocytosis needs to be addressed.
Response3: Indeed, ceramides have been implicated in a large variety of cellular functions including but not limited to cell proliferation, cell death, migration and chemotaxis, cell signaling and membrane repair. We absolutely agree that the probes will be useful for studies beyond the role of ceramide in phagocytosis, but we argue that this would be best achieved by groups with deeper expertise in these domains of study than ourselves. We plan to make the probes available to anyone who is interested in working with them by sharing through the plasmid repository Addgene. We now briefly mention the potential use of the probes for investigating plasma membrane repair and ceramide-induced cell death in Pg 17 lines 516-518.
Comment4: Specificity and Selectivity of Probes: The manuscript might need a more detailed evaluation of the specificity and selectivity of the probes for ceramide versus other sphingolipids or cellular components, which is crucial for their effective application.
Response4: We absolutely agree that this is a key point that must be addressed, but it is not trivial to achieve. For most lipid probes, such as the phosphoinositide probes, specificity and selectivity have been defined using in vitro binding studies with purified probes. However, for many sphingolipids, in vitro studies are notoriously challenging, one of the major reasons being that ceramides and sphingosines themselves will deform or permeabilize membranes and are thus not very stable in model membranes. This is perhaps one of the reasons why our understanding of sphingolipids has lagged behind that of other lipid classes. Indeed, as nicely summarized in Canal et al 2018, many hypotheses of the mechanisms underlying how ceramides act involve aggregation or coalescence to form channels or “pinching off” of microvesicles in processes that are not driven by proteins, but rather by the physical nature of the lipids themselves. We did try several different in vitro methods including liposome flotation and fat blot assays to get alternative read-outs of binding specificity, and even the liposome microarrays were performed many many times with nearly one thousand liposome spots screened. Yet still for some liposome compositions only one or two spots were successful, and for the other methods we could not get consistent results. Adding to this the fact that the purified KSR probes themselves tended to aggregate and precipitate in solution, we believe that in vitro studies are not fruitful, and that further analysis of selectivity and specificity will need to be performed in the context of living cells in future studies dedicated to the task. We now discuss this in greater detail in Pgs 14-15 lines 408-527 and provide suggestions for how this may be achieved.
Comment5: Minor Concerns: Concerns regarding the stability of the probes under different experimental conditions and their resistance to photobleaching, which is essential for their utility in long-term imaging studies.
Response5: We now include a more extended discussion on how the stability of the probes must be examined in further details, giving examples of how this might be achieved. We also mention that defining the long-term stability of the probe will be critical before performing long-term imaging studies. (Pgs 14-15, Lines 445-450)
Comment6: Potential interference of the probes with normal cellular processes, including whether their presence affects the natural dynamics of ceramide or cellular responses during phagocytosis.
Response6: Indeed, this caveat, which is a concern for all fluorescent lipid probes, was also brought up by Reviewer 1. We have now added a new lipidomic analysis that shows that, at least for C-KSR expression, the disruption to normal sphingolipid metabolism is not major (new Fig S2A and new Supplementary Data S1). However, obviously triggering autophagy with this probe is an effect that must be considered, and we do suggest that in the future it would be interesting to re-design a KSR-based construct that has a high turnover rate, for example by adding a proteasome-targeting sequence instead, which might be less disruptive than triggering autophagy. Furthermore, this might help with some of the toxicity observed for C-KSR-GS. We cannot exclude at this point however that N-KSR or C-KSR-GS do not alter normal sphingolipid metabolism or other cellular processes, and we now discuss this more directly within the results (Pg 12, Lines 380-387) and discussion (Pgs 15-16, Lines 485-506).
Comment7: The study could benefit from a more robust quantitative analysis of the fluorescence changes, including calibration methods to correlate fluorescence intensity with ceramide concentrations.
Response7: We absolutely agree that quantification of fluorescent changes, though challenging, are very important and we now include a quantification of the fluorescent changes during phagocytosis in the new Figure 6A-C. In these experiments the blind quantification of C-KSR recruitment to the plasma membrane correlated well (increase at 30 min and decrease back to baseline at 135 min) with the time-dependent changes in total cellular ceramide concentrations measured by mass spectrometry. Whereas C-KSR-GS behaved similarly at 30 min, the trend towards a decrease at 135 min was not significant. However, notwithstanding the difficulty in reliable image segmentation because of the accumulating beads at this time point, both N-KSR and C-KSR-GS also displayed a trends towards a concomitant time-dependent decrease in global cellular fluorescence, indicating a more complex behavior of these two probes. These new results and their implications are highlighted in the revised results section (Pg 12 Lines 313-387).
For the calibration, the idea is really intriguing, but difficult to conduct properly experimentally. The minimal requirement for such a calibration would be to deplete or increase cellular ceramides to obtain maximum and minimum values and compare this to baseline fluorescence levels. Since cellular ceramide depletion is very slow (on the order of several hours) even in conditions of maximum inhibition of the biosynthetic pathway, and for a maximum signal even the soluble C6-ceramide takes some time to be internalized and produce a change in fluorescence, it is impractical (and more than likely lethal) to induce these changes in the same set of cells as one might do, for example, for a calcium probe calibration. Thus, our data on myriocin and C6-ceramide already provide a rough minimum and maximum range, as higher concentrations of C6-ceramide were lethal. Another method might be to perform a time-dependent fluorescent measurement using a concentration curve of C6-ceramide, i.e. in a 96-well plate-reader format. However, since C6-ceramide is rapidly metabolized, such a measurement would have little true meaning without accompanying lipidomic data to gauge the true levels of ceramide under these conditions. As such, we feel the experimentation required is outside the scope of the current study but would certainly be interesting to perform in the future. We now mention the need for future studies to address calibration (Pg. 15 Lines 481-484) and thank the reviewer for this interesting suggestion.
Comment8: A comparative analysis with existing methods for tracking ceramide dynamics, to highlight the advantages or limitations of the proposed probes.
Response8: To address the important point raised by the reviewer we did attempt antibody stainings with the commercially available anti-ceramide antibody (MID 15B4 from Enzo). However, in our hands the staining remained diffuse, looking non-specific (see example in attached file), and did not change with myriocin (or fumonisin b1) treatment, despite several different fixation (PFA 4%, 1%) and permeabilization protocols (Triton 0.1%,0.2% 0.5%, saponin 0.1%, 0.05%, freeze-thaw). Since mass spectrometry (and changes in growth and physical morphology) confirmed the efficiency of the drugs treatment on reducing ceramides under similar conditions, we are confident that our drug treatments worked. However, since the antibody is cited in numerous (>40) studies, and we cannot be sure that alternative staining protocols (or perhaps the use of a different cell type) might not eventually prove successful we feel it is best not to include these negative data in the current manuscript.
We have now however extended our discussion of existing methods for tracking ceramides in the introduction (Pg 2, Lines 51 to 63) and discussion (Pg 16, Lines 507 to 512) to further highlight the need for new methods that complement existing ones to track this elusive lipid.
Comment9: Discussion of potential artifacts in the imaging process and limitations of the probes, including their potential to aggregate or mislocalize within the cell.
Response9: This is now briefly discussed in Pg 15 Lines 498-506.
Comment10: Addressing these concerns would provide a more comprehensive evaluation of the probes' capabilities and limitations, enhancing the manuscript's overall scientific rigor and impact.
Response10: We thank the reviewer for these insightful suggestions which we believe have helped us improve the relevance and utility of our manuscript.

Round 2
Reviewer 1 Report
Comments and Suggestions for Authors
I thank the authors for thorough revising the manuscript and providing commentaries to my questions. I think the manuscript and supplementary materials have been significantly improved and became much more readable and understandable. I believe now the manuscript can be accepted for the publication in IJMS.